# Pre-Training Protein Encoder via Siamese Sequence-Structure Diffusion Trajectory Prediction

**Zuobai Zhang**[1,2][*]**, Minghao Xu**[1,2][*]**, Aurélie Lozano**[3]**,**
**Vijil Chenthamarakshan**[3]**, Payel Das**[3][†]**, Jian Tang**[1,4,5][†]

[*]equal contribution      [†]corresponding author
[1]Mila - Québec AI Institute, [2]Université de Montréal, [3]IBM Research,
[4]HEC Montréal, [5]CIFAR AI Chair
{zuobai.zhang, minghao.xu}@mila.quebec,
{ecvijil,aclozano,daspa}@us.ibm.com, jian.tang@hec.ca

## Abstract

Self-supervised pre-training methods on proteins have recently gained attention, with most approaches focusing on either protein sequences or structures, neglecting the exploration of their joint distribution, which is crucial for a comprehensive understanding of protein functions by integrating co-evolutionary information and structural characteristics. In this work, inspired by the success of denoising diffusion models in generative tasks, we propose the **DiffPreT** approach to pre-train a protein encoder by sequence-structure joint diffusion modeling. DiffPreT guides the encoder to recover the native protein sequences and structures from the perturbed ones along the joint diffusion trajectory, which acquires the joint distribution of sequences and structures. Considering the essential protein conformational variations, we enhance DiffPreT by a method called Siamese Diffusion Trajectory Prediction (**SiamDiff**) to capture the correlation between different conformers of a protein. SiamDiff attains this goal by maximizing the mutual information between representations of diffusion trajectories of structurally-correlated conformers. We study the effectiveness of DiffPreT and SiamDiff on both atom- and residue-level structure-based protein understanding tasks. Experimental results show that the performance of DiffPreT is consistently competitive on all tasks, and SiamDiff achieves new state-of-the-art performance, considering the mean ranks on all tasks. Code will be released upon acceptance.

## 1 Introduction

Machine learning-based methods have made remarkable strides in predicting protein structures [44, 5, 50] and functionality [55, 24]. Among them, self-supervised (unsupervised) pre-training approaches [20, 61, 87] have been successful in learning effective protein representations from available protein sequences or from their experimental/predicted structures. These pre-training approaches are based on the rationale that modeling the input distribution of proteins provides favorable initialization of model parameters and serves as effective regularization for downstream tasks [25]. Previous methods have primarily focused on modeling the marginal distribution of either protein sequences to acquire co-evolutionary information [20, 61], or protein structures to capture essential characteristics for tasks such as function prediction and fold classification [87, 33]. Nevertheless, both these forms of information hold significance in revealing the underlying functions of proteins and offer complementary perspectives that are still not extensively explored. To address this gap, a more promising approach for pre-training could involve modeling the joint distribution of protein sequences and structures, surpassing the limitations of unimodal pre-training methods.

To model this joint distribution, denoising diffusion models [37, 65] have recently emerged as one of the most effective methods due to their simple training objective and high sampling quality and diversity [2, 54, 40, 74]. However, the application of diffusion models has predominantly been explored in the context of generative tasks, rather than within pre-training and fine-tuning frameworks that aim to learn effective representations for downstream tasks. In this work, we present a novel approach that adapts denoising diffusion models to pre-train structure-informed protein encoders[1]. Our proposed approach, called **DiffPreT**, gradually adds noise to both protein sequence and structure to transform them towards random distribution, and then denoises the corrupted protein structure and sequence using a noise prediction network parameterized with the output of the protein encoder. This approach enables the encoder to learn informative representations that capture the inter-atomic interactions within the protein structure, the residue type dependencies along the protein sequence, and the joint effect of sequence and structure variations.

In spite of these advantages, DiffPreT ignores the fact that any protein structure exists as a population of interconverting conformers, and elucidating this conformational heterogeneity is essential for predicting protein function and ligand binding [27]. In both DiffPreT and previous studies, no explicit constraints are added to acquire the structural correlation between different conformers of a specific protein or between structural homologs, which prohibits capturing the conformational energy landscape of a protein [56]. Therefore, to consider the physics underlying the conformational change, we propose Siamese Diffusion Trajectory Prediction (**SiamDiff**) to augment the DiffPreT by maximizing the mutual information between representations of diffusion trajectories of structurally-correlated conformers (*i.e.*, siamese diffusion trajectories). We first adopt a torsional perturbation scheme on the side chain to generate randomly simulated conformers [36]. Then, for each protein, we generate diffusion trajectories for a pair of its conformers. We theoretically prove that the problem can be transformed to the mutual prediction of the trajectories using representations from their counterparts. In this way, the model can keep the advantages of DiffPreT and inject conformer-related information into representations as well.

Both DiffPreT and SiamDiff can be flexibly applied to atom-level and residue-level structures to pre-train protein representations. To thoroughly assess their capabilities, we conduct extensive evaluations of the pre-trained models on a wide range of downstream protein understanding tasks. These tasks encompass protein function annotation, protein-protein interaction prediction, mutational effect prediction, residue structural contributions, and protein structure ranking. In comparison to existing pre-training methods that typically excel in only a subset of the considered tasks, DiffPreT consistently delivers competitive performance across all tasks and at both the atomic and residue-level resolutions. Moreover, SiamDiff further enhances model performance, surpassing previous state-of-the-art results in terms of mean ranks across all evaluated tasks.

## 2 DiffPreT: Diffusion Models for Pre-Training

Recently, there have been promising progress on applying denoising diffusion models for protein structure-sequence co-design [54, 2]. The effectiveness of the *joint diffusion model* on modeling the distribution of proteins suggests that the process may reflect physical and chemical principles underlying protein formation [3, 18], which could be beneficial for learning informative representations. Based on this intuition, in this section, we explore the application of joint diffusion models on pre-training protein encoders in a pre-training and fine-tuning framework.

### 2.1 Preliminary

**Notation.** A protein with $n_r$ residues (amino acids) and $n_a$ atoms can be represented as a sequence-structure tuple $\mathcal{P} = (\mathcal{S}, \mathcal{R})$. We use $\mathcal{S} = [s_1, s_2, \cdots, s_{n_r}]$ to denote its sequence with $s_i$ as the type of the $i$-th residue, while $\mathcal{R} = [r_1, r_2..., r_{n_a}] \in \mathbb{R}^{n_a \times 3}$ denotes its structure with $r_i$ as the Cartesian coordinates of the $i$-th atom. To model the structure, we construct a graph for each protein with edges connecting atoms whose Euclidean distance below a certain threshold.

**Equivariance.** *Equivariance* is ubiquitous in machine learning for modeling the symmetry in physical systems [67, 75] and is shown to be critical for successful design and better generalization of 3D

---

[1]It is important to note that protein structure encoders in this context take both sequences and structures as input, distinguishing them from protein sequence encoders as established in previous works.

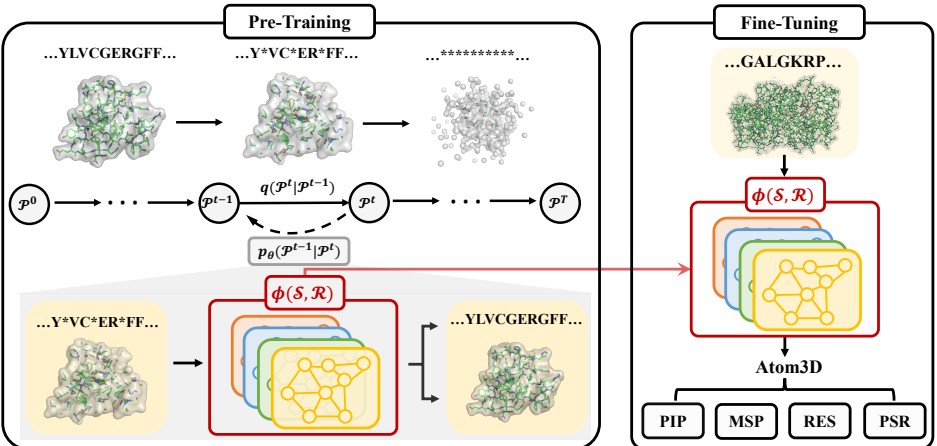

Figure 1: A pre-training and fine-tuning framework for DiffPreT. During pre-training, diffusion processes are applied to both protein structures and sequences, with * indicating masked residues. Noise prediction networks, parameterized with an encoder $\phi(\mathcal{S}, \mathcal{R})$, are employed to restore the original states. The learned encoder $\phi(\mathcal{S}, \mathcal{R})$ is subsequently fine-tuned on downstream tasks.

networks [46]. Formally, a function $\mathcal{F} : \mathcal{X} \rightarrow \mathcal{Y}$ is equivariant *w.r.t.* a group $G$ if $\mathcal{F} \circ \rho_{\mathcal{X}}(x) = \rho_{\mathcal{Y}} \circ \mathcal{F}(x)$, where $\rho_{\mathcal{X}}$ and $\rho_{\mathcal{Y}}$ are transformations corresponding to an element $g \in G$ acting on the space $\mathcal{X}$ and $\mathcal{Y}$, respectively. The function is invariant *w.r.t* $G$ if the transformations $\rho_{\mathcal{Y}}$ is identity. In this paper, we consider SE(3) group, *i.e.*, rotations and translations in 3D space.

**Problem Definition.** Given a set of unlabeled proteins $\mathcal{D} = \{\mathcal{P}_1, \mathcal{P}_2, ...\}$, our goal is to train a protein encoder $\phi(\mathcal{S}, \mathcal{R})$ to extract informative $d$-dimensional residue representations $\boldsymbol{h} \in \mathbb{R}^{n_r \times d}$ and atom representations $\boldsymbol{a} \in \mathbb{R}^{n_a \times d}$ that are SE(3)-invariant *w.r.t.* protein structures $\mathcal{R}$.

## 2.2 Diffusion Models on Proteins

Diffusion models are a class of deep generative models with latent variables encoded by a *forward diffusion process* and decoded by *a reverse generative process* [63]. We use $\mathcal{P}^0$ to denote the ground-truth protein and $\mathcal{P}^t$ for $t = 1, \cdots, T$ to be the latent variables over $T$ diffusion steps. Modeling the protein as an evolving thermodynamic system, the forward process gradually injects small noise to the data $\mathcal{P}^0$ until reaching a random noise distribution at time $T$. The reverse process learns to denoise the latent variable towards the data distribution. Both processes are defined as Markov chains:

$$q(\mathcal{P}^{1:T}|\mathcal{P}^0) = \prod_{t=1}^{T} q(\mathcal{P}^t|\mathcal{P}^{t-1}), \quad p_\theta(\mathcal{P}^{0:T-1}|\mathcal{P}^T) = \prod_{t=1}^{T} p_\theta(\mathcal{P}^{t-1}|\mathcal{P}^t), \quad (1)$$

where $q(\mathcal{P}^t|\mathcal{P}^{t-1})$ defines the forward process at step $t$ and $p_\theta(\mathcal{P}^{t-1}|\mathcal{P}^t)$ with learnable parameters $\theta$ defines the reverse process at step $t$. We decompose the forward process into diffusion on protein structures and sequences, respectively. The decomposition is represented as follows:

$$q(\mathcal{P}^t|\mathcal{P}^{t-1}) = q(\mathcal{R}^t|\mathcal{R}^{t-1}) \cdot q(\mathcal{S}^t|\mathcal{S}^{t-1}), \quad p_\theta(\mathcal{P}^{t-1}|\mathcal{P}^t) = p_\theta(\mathcal{R}^{t-1}|\mathcal{P}^t) \cdot p_\theta(\mathcal{S}^{t-1}|\mathcal{P}^t), \quad (2)$$

where the reverse processes use representations $\boldsymbol{a}^t$ and $\boldsymbol{h}^t$ learned by the protein encoder $\phi_\theta(\mathcal{S}^t, \mathcal{R}^t)$.

**Forward diffusion process** $q(\mathcal{P}^t|\mathcal{P}^{t-1})$. For diffusion on protein structures, we introduce random Gaussian noises to the 3D coordinates of the structure. For diffusion on sequences, we utilize a Markov chain approach with an absorbing state [MASK], where each residue either remains the same or transitions to [MASK] with a certain probability at each time step [4]. Specifically, we have:

$$q(\mathcal{R}^t|\mathcal{R}^{t-1}) = \mathcal{N}(\mathcal{R}^t; \sqrt{1 - \beta_t}\mathcal{R}^{t-1}, \beta_t I), \quad q(\mathcal{S}^t|\mathcal{S}^{t-1}) = \text{random\_mask}(\mathcal{S}^{t-1}, \rho_t), \quad (3)$$

Here, $\beta_1, ..., \beta_T$ and $\rho_1, ..., \rho_T$ are a series of fixed variances and masking ratios, respectively. random_mask$(\mathcal{S}^{t-1}, \rho_t)$ denotes the random masking operation, where each residue in $\mathcal{S}^{t-1}$ is masked with a probability of $\rho_t$ at time step $t$.

**Reverse process on structures** $p_\theta(\mathcal{R}^{t-1}|\mathcal{P}^t)$. The reverse process on structures is parameterized as a Gaussian with a learnable mean $\mu_\theta(\mathcal{P}^t, t)$ and user-defined variance $\sigma_t$. Given the availability of

$\mathcal{R}^t$ as an input, we reparameterize the mean $\mu_\theta(\mathcal{P}^t, t)$ following Ho et al. [37]:

$$p_\theta(\mathcal{R}^{t-1}|\mathcal{P}^t) = \mathcal{N}(\mathcal{R}^{t-1}; \mu_\theta(\mathcal{P}^t, t), \sigma_t^2 I), \quad \mu_\theta(\mathcal{P}^t, t) = \frac{1}{\sqrt{\alpha_t}}\left(\mathcal{R}^t - \frac{\beta_t}{\sqrt{1-\bar{\alpha}_t}}\epsilon_\theta(\mathcal{P}^t, t)\right), \quad (4)$$

where $\alpha_t = 1 - \beta_t$, $\bar{\alpha}_t = \prod_{s=1}^{t} \alpha_s$ and the network $\epsilon_\theta(\cdot)$ learns to decorrupt the data and should be translation-invariant and rotation-equivariant *w.r.t.* the structure $\mathcal{R}^t$.

To define our noise prediction network, we utilize the atom representations $\boldsymbol{a}^t$ (which is guaranteed to be SE(3)-invariant *w.r.t.* $\mathcal{R}^t$ by the encoder) and atom coordinates $\boldsymbol{r}^t$ (which is SE(3)-equivariant *w.r.t.* $\mathcal{R}^t$). We build an equivariant output based on normalized directional vectors between adjacent atom pairs. Each edge $(i, j)$ is encoded by its length $\|\boldsymbol{r}_i^t - \boldsymbol{r}_j^t\|_2$ and the representations of two end nodes $\boldsymbol{a}_i^t$, $\boldsymbol{a}_j^t$, and the encoded score $m_{i,j}$ will be used for aggregating directional vectors. Specifically,

$$[\epsilon_\theta(\mathcal{P}^t, t)]_i = \sum_{j\in\mathcal{N}^t(i)} m_{i,j} \cdot \frac{\boldsymbol{r}_i^t - \boldsymbol{r}_j^t}{\|\boldsymbol{r}_i^t - \boldsymbol{r}_j^t\|_2}, \quad \text{with } m_{i,j} = \text{MLP}(\boldsymbol{a}_i^t, \boldsymbol{a}_j^t, \text{MLP}(\|\boldsymbol{r}_i^t - \boldsymbol{r}_j^t\|_2)), \quad (5)$$

where $\mathcal{N}^t(i)$ denotes the neighbors of the atom $i$ in the corresponding graph of $\mathcal{P}^t$. Note that $\epsilon_\theta(\mathcal{P}^t, t)$ achieves the equivariance requirement, as $m_{i,j}$ is SE(3)-invariant *w.r.t.* $\mathcal{R}^t$ while $\boldsymbol{r}_i^t - \boldsymbol{r}_j^t$ is translation-invariant and rotation-equivariant *w.r.t.* $\mathcal{R}^t$.

**Reverse process on sequences** $p_\theta(\mathcal{S}^{t-1}|\mathcal{P}^t)$. For the reverse process $p_\theta(\mathcal{S}^{t-1}|\mathcal{P}^t)$, we adopt the parameterization proposed in [4]. The diffusion trajectory is characterized by the probability $q(\mathcal{S}^{t-1}|\mathcal{S}^t, \tilde{\mathcal{S}}^0)$, and we employ a network $\tilde{p}\theta$ to predict the probability of $\mathcal{S}^0$:

$$p_\theta(\mathcal{S}^{t-1}|\mathcal{P}^t) \propto \sum_{\tilde{\mathcal{S}}^0} q(\mathcal{S}^{t-1}|\mathcal{S}^t, \tilde{\mathcal{S}}^0) \cdot \tilde{p}_\theta(\tilde{\mathcal{S}}^0|\mathcal{P}^t), \quad (6)$$

We define the predictor $\tilde{p}_\theta$ with residue representations $\boldsymbol{h}^t$. For each masked residue $i$ in $\mathcal{S}^t$, we feed its representation $\boldsymbol{h}_i^t$ to an MLP and predict the type of the corresponding residue type $s_i^0$ in $\mathcal{S}^0$:

$$\tilde{p}_\theta(\tilde{\mathcal{S}}^0|\mathcal{P}^t) = \prod_i \tilde{p}_\theta(\tilde{s}_i^0|\mathcal{P}^t) = \prod_i \text{Softmax}(\tilde{s}_i^0|\text{MLP}(\boldsymbol{h}_i^t)), \quad (7)$$

where the softmax function is applied over all residue types.

## 2.3 Pre-Training Objective

Now we derive the pre-training objective of DiffPreT by optimizing the diffusion model above with the ELBO loss [37]:

$$\mathcal{L} := \mathbb{E}\left[\sum_{t=1}^{T} D_{\text{KL}}\left(q(\mathcal{P}^{t-1}|\mathcal{P}^t, \mathcal{P}^0)\|p_\theta(\mathcal{P}^{t-1}|\mathcal{P}^t)\right)\right]. \quad (8)$$

Under the assumptions in (2), it can be shown that the objective can be decomposed into a structure loss $\mathcal{L}^{(\mathcal{R})}$ and a sequence loss $\mathcal{L}^{(\mathcal{S})}$ (see proof in App. C.2):

$$\begin{aligned}
\mathcal{L}^{(\mathcal{R})} &:= \mathbb{E}\left[\sum_{t=1}^{T} D_{\text{KL}}\left(q(\mathcal{R}^{t-1}|\mathcal{R}^t, \mathcal{R}^0)\|p_\theta(\mathcal{R}^{t-1}|\mathcal{P}^t)\right)\right], \\
\mathcal{L}^{(\mathcal{S})} &:= \mathbb{E}\left[\sum_{t=1}^{T} D_{\text{KL}}\left(q(\mathcal{S}^{t-1}|\mathcal{S}^t, \mathcal{S}^0)\|p_\theta(\mathcal{S}^{t-1}|\mathcal{P}^t)\right)\right].
\end{aligned} \quad (9)$$

Both loss functions can be simplified as follows.

**Structure loss** $\mathcal{L}^{(\mathcal{R})}$. It has been shown in Ho et al. [37] that the loss function can be simplified under our parameterization by calculating KL divergence between Gaussians as weighted L2 distances between means $\epsilon_\theta$ and $\epsilon$ (see details in App. C.3):

$$\mathcal{L}^{(\mathcal{R})} = \sum_{t=1}^{T} \gamma_t \mathbb{E}_{\epsilon\sim\mathcal{N}(0,I)}\left[\|\epsilon - \epsilon_\theta(\mathcal{P}^t, t)\|_2^2\right], \quad (10)$$

where the coefficients $\gamma_t$ are determined by the variances $\beta_1, ..., \beta_t$. In practice, we follow Ho et al. [37] to set all weights $\gamma_t = 1$ for the simplified loss $\mathcal{L}^{(\mathcal{R})}_{\text{simple}}$.

Since $\epsilon_\theta$ is designed to be rotation-equivariant *w.r.t.* $\mathcal{R}^t$, to make the loss function invariant *w.r.t.* $\mathcal{R}^t$, the supervision $\epsilon$ is also supposed to achieve such equivariance. Therefore, we adopt the chain-rule approach proposed in Xu et al. [82], which decomposes the noise on pairwise distances to obtain the modified noise vector $\hat{\epsilon}$ as supervision. We refer readers to Xu et al. [82] for more details.

**Sequence loss** $\mathcal{L}^{(\mathcal{S})}$. Since we parameterize $p_\theta(\mathcal{S}^{t-1}|\mathcal{P}^t)$ with $\tilde{p}_\theta(\tilde{\mathcal{S}}^0|\mathcal{P}^t)$ and $q(\mathcal{S}^{t-1}|\mathcal{S}^t, \tilde{\mathcal{S}}^0)$ as in (6), it can be proven that the $t$-th KL divergence term in $\mathcal{L}^{(\mathcal{S})}$ reaches zero when $\tilde{p}_\theta(\tilde{\mathcal{S}}^0|\mathcal{P}^t)$ assigns

all mass on the ground truth $\mathcal{S}^0$ (see proof in App. C.4). Therefore, for pre-training, we can simplify the KL divergence to the cross-entropy between the correct residue type $s_i^0$ and the prediction:

$$\mathcal{L}_{\text{simple}}^{(\mathcal{S})} = \sum_{t=1}^{T} \sum_i \text{CE}\left(s_i^0, \tilde{p}_\theta(s_i^0 | \mathcal{P}^t)\right), \tag{11}$$

where $\text{CE}(\cdot, \cdot)$ denotes the cross-entropy loss.

The ultimate training objective is the sum of simplified structure and sequence diffusion losses:

$$\mathcal{L}_{\text{simple}} = \mathcal{L}_{\text{simple}}^{(\mathcal{R})} + \mathcal{L}_{\text{simple}}^{(\mathcal{S})}. \tag{12}$$

### 2.4 Two-Stage Noise Scheduling

Previous studies on scheduled denoising autoencoders on images [23] have shown that large noise levels encourage the learning of coarse-grained features, while small noise levels require the model to learn fine-grained features. We observe a similar phenomenon in structure diffusion, as depicted in Fig. 2. The diffusion loss with large noise at a fixed scale (orange) is smaller than that with small noise at a fixed scale (green), indicating the higher difficulty of structure diffusion with small noises. Interestingly, the opposite is observed in sequence diffusion, where the denoising accuracy with small noise (green) is higher than that with large noise (orange). This can be attributed to the joint diffusion effects on protein sequences and structures. The addition of large noise during joint diffusion significantly disrupts protein structures, making it more challenging to infer the correct protein sequences. For validation, we com-

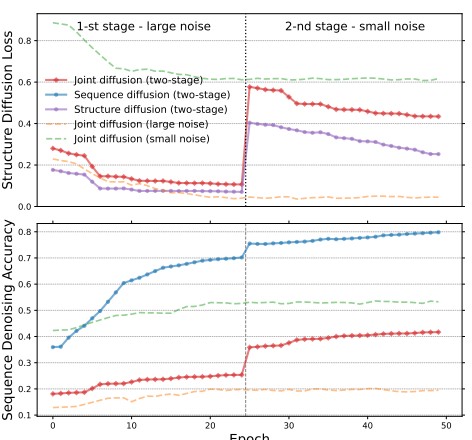

Figure 2: Structure diffusion loss and sequence denoising accuracy of pre-training with different noise scheduling strategies.

pare the loss of sequence-only diffusion pre-training (blue) with joint diffusion pre-training (red) in Fig. 2. Sequence diffusion achieves higher denoising accuracy than joint diffusion when using uncorrupted structures, supporting our hypothesis.

Unlike recent denoising pre-training methods that rely on a fixed noise scale as a hyperparameter [85], DiffPreT incorporates a denoising objective at various noise levels to capture both coarse- and fine-grained features and consider the joint diffusion effect explained earlier. Both granularities of features are crucial for downstream tasks, as they capture both small modifications for assessing protein structure quality and large changes leading to structural instability. In our implementation, we perform diffusion pre-training using $T = 100$ noise levels (time steps). Following the intuition of learning coarse-grained features before fine-grained ones in curriculum learning [8], we divide the pre-training process into two stages: the first stage focuses on large noise levels ($t = 10, ..., 100$), while the second stage targets small noise levels ($t = 1, ..., 9$). As observed in Fig. 2, structure diffusion becomes more challenging and sequence diffusion becomes easier during the second stage, as we discussed earlier. However, even with this two-stage diffusion strategy, there remains a significant gap between the accuracy of sequence diffusion and joint diffusion. We hypothesize that employing protein encoders with larger capacities could help narrow this gap, which is left as our future works.

## 3 SiamDiff: Siamese Diffusion Trajectory Prediction

Through diffusion models on both protein sequences and structures, the pre-training approach proposed in Sec. 2 tries to make representations capture (1) the atom- and residue-level spatial interactions and (2) the statistical dependencies of residue types within a single protein. Nevertheless, no constraints or supervision have been added for modeling relations between different protein structures, especially different conformers of the same protein. Generated under different environmental factors, these conformers typically share the same protein sequence but different structures due to side chain rotation driving conformational variations. These conformers' properties are highly correlated [58], and their representations should reflect this correlation.

In this section, we introduce Siamese Diffusion Trajectory Prediction (**SiamDiff**), which incorporates conformer-related information into DiffPreT by maximizing mutual information (MI) between diffusion trajectory representations of correlated conformers. We propose a scheme to generate simulated conformers (Sec. 3.1), generate joint diffusion trajectories (Sec. 3.2), and transform MI maximization into mutual denoising between trajectories (Sec. 3.3), sharing similar loss with DiffPreT.

### 3.1 Conformer Simulation Scheme

Our method begins by generating different conformers of a given protein. However, direct sampling requires an accurate characterization of the energy landscape of protein conformations, which can be difficult and time-consuming. To address this issue, we adopt a commonly used scheme for sampling randomly simulated conformers by adding torsional perturbations to the side chains [36].

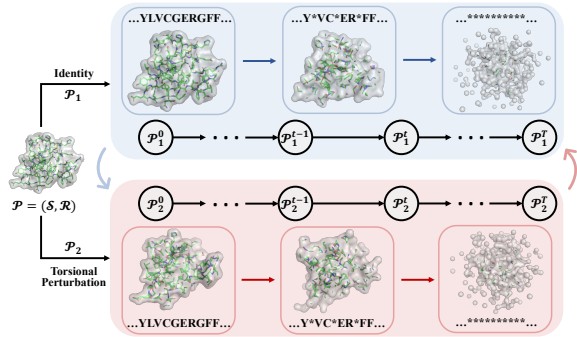

Specifically, given the original protein $\mathcal{P} = (\mathcal{S}, \mathcal{R})$, we consider it as the native state $\mathcal{P}_1 = (\mathcal{S}_1, \mathcal{R}_1)$ and generate a correlated conformer $\mathcal{P}_2 = (\mathcal{S}_2, \mathcal{R}_2)$ by

Figure 3: High-level illustration of SiamDiff. Mutual denoising of diffusion trajectories is performed across two correlated conformers $\mathcal{P}_1$ and $\mathcal{P}_2$.

randomly perturbing the protein structure. That is, we set $\mathcal{S}_2$ to be the same as $\mathcal{S}_1$, and $\mathcal{R}_2$ is obtained by applying a perturbation function perturb($\mathcal{R}_1, \epsilon$) to the original residue structure, where $\epsilon \in [0, 2\pi)^{n_r \times 4}$ is a noise vector drawn from a wrapped normal distribution [15]. The perturb($\cdot, \cdot$) function rotates the side-chain of each residue according to the sampled torsional noises. To avoid atom clashes, we adjust the variance of the added noise and regenerate any conformers that violate physical constraints. It should be noted that the scheme can be adapted flexibly when considering different granularities of structures. For example, instead of considering side chain rotation on a fixed backbone, we can also consider a flexible backbone by rotating backbone angles, thereby generating approximate conformers.

Although the current torsional perturbation scheme is effective, there is potential for improvement. Future research can explore enhancements such as incorporating available rotamer libraries [62] and introducing backbone flexibility using empirical force fields [72].

### 3.2 Siamese Diffusion Trajectory Generation

To maintain the information-rich joint diffusion trajectories from DiffPreT, we generate trajectories for pairs of conformers, *a.k.a.*, siamese trajectories. We first sample the diffusion trajectories $\mathcal{P}_1^{0:T}$ and $\mathcal{P}_2^{0:T}$ for conformers $\mathcal{P}_1$ and $\mathcal{P}_2$, respectively, using the joint diffusion process outlined in Sec. 2. For example, starting from $\mathcal{P}_1^0 = \mathcal{P}_1 = (\mathcal{S}_1, \mathcal{R}_1)$, we use the joint diffusion on structures and sequences to define the diffusion process $q(\mathcal{P}_1^{1:T}|\mathcal{P}_1^0) = q(\mathcal{R}_1^{1:T}|\mathcal{R}_1^0)q(\mathcal{S}_1^{1:T}|\mathcal{S}_1^0)$. We derive trajectories on structures $\mathcal{R}_1^{1:T}$ using the Gaussian noise in (3) and derive the sequence diffusion process $\mathcal{S}^{1:T}$ using the random masking in (6). In this way, we define the trajectory $\mathcal{P}_1^{0:T} = \{(\mathcal{S}_1^t, \mathcal{R}_1^t)\}_{t=0}^T$ for $\mathcal{P}_1$ and can derive the siamese trajectory $\mathcal{P}_2^{0:T} = \{(\mathcal{S}_2^t, \mathcal{R}_2^t)\}_{t=0}^T$ similarly.

### 3.3 Mutual Information Maximization between Representations of Siamese Trajectories

We aim to maximize the mutual information (MI) between representations of siamese trajectories constructed in a way that reflects the correlation between different conformers of the same protein. Direct optimization of MI is intractable, so we instead maximize a lower bound. In App. C.1, we show that this problem can be transformed into the minimization of a loss function for mutual denoising between two trajectories: $\mathcal{L} = \frac{1}{2}(\mathcal{L}^{(2 \rightarrow 1)} + \mathcal{L}^{(1 \rightarrow 2)})$, where

$$\mathcal{L}^{(b \rightarrow a)} := \mathbb{E}_{\mathcal{P}_a^{0:T}, \mathcal{P}_b^{0:T}} \left[ \sum_{t=1}^T D_{\text{KL}} \left( q(\mathcal{P}_a^{t-1}|\mathcal{P}_a^t, \mathcal{P}_a^0) || p(\mathcal{P}_a^{t-1}|\mathcal{P}_a^t, \boldsymbol{\mathcal{P}}_b^{0:T}) \right) \right], \tag{13}$$

with $b \rightarrow a$ being either $2 \rightarrow 1$ or $1 \rightarrow 2$ and $\boldsymbol{\mathcal{P}}_b^{0:T}$ being representations of the trajectory $\mathcal{P}_b^{0:T}$.

The two terms share the similar formula as the ELBO loss in (8). Take $\mathcal{L}^{(2\to1)}$ for example. Here $q(\mathcal{P}_1^{t-1}|\mathcal{P}_1^t,\mathcal{P}_1^0)$ is a posterior analytically tractable with our definition of each diffusion step $q(\mathcal{P}_1^t|\mathcal{P}_1^{t-1})$ in (3) and (6). The reverse process is learnt to generate a less noisy state $\mathcal{P}_1^{t-1}$ given the current state $\mathcal{P}_1^t$ and representations of the siamese trajectory $\boldsymbol{\mathcal{P}}_2^{0:T}$, which are extracted by the protein encoder to be pre-trained. The parameterization of the reverse process is similar as in Sec. 2.2, with the representations replaced by those of $\boldsymbol{\mathcal{P}}_2^{0:T}$ (see App. B for details).

Our approach involves mutual prediction between two siamese trajectories, which is similar to the idea of mutual representation reconstruction in [28, 14]. However, since $\mathcal{P}_1$ and $\mathcal{P}_2$ share information about the same protein, the whole trajectory of $\mathcal{P}_2$ could provide too many clues for denoising towards $\mathcal{P}_1^{t-1}$, making the pre-training task trivial. To address this issue, we parameterize $p(\mathcal{P}_1^{t-1}|\mathcal{P}_1^t,\boldsymbol{\mathcal{P}}_2^{0:T})$ with $p_\theta(\mathcal{P}_1^{t-1}|\mathcal{P}_1^t,\boldsymbol{\mathcal{P}}_2^t)$. For diffusion on sequences, we further guarantee that the same set of residues are masked in $\mathcal{S}_1^t$ and $\mathcal{S}_2^t$ to avoid leakage of ground-truth residue types across correlated trajectories.

**Final pre-training objective.** Given the similarity between the one-side objective and the ELBO loss in (8), we can use a similar way to decompose the objective into structure and sequence losses and then derive simplified loss functions for each side. To summarize, the ultimate training objective for our method is

$$\mathcal{L}_{\text{simple}} = \tfrac{1}{2}(\mathcal{L}_{\text{simple}}^{(\mathcal{R},2\to1)} + \mathcal{L}_{\text{simple}}^{(\mathcal{S},2\to1)} + \mathcal{L}_{\text{simple}}^{(\mathcal{R},1\to2)} + \mathcal{L}_{\text{simple}}^{(\mathcal{S},1\to2)}), \tag{14}$$

where $\mathcal{L}_{\text{simple}}^{(\cdot,b\to a)}$ is the loss term defined by predicting $\mathcal{P}_a^{0:T}$ from $\mathcal{P}_b^{0:T}$ (see App. B for derivation).

### 3.4 Residue-level model

Residue-level protein graphs are simplified atom graphs that enable efficient message passing between nodes and edges. As in Zhang et al. [87], we only keep the alpha carbon atom of each residue and add sequential, radius and K-nearest neighbor edges as different types of edges. For SiamDiff, the residue-level model cannot discriminate conformers generated by rotating side chains, since we only keep CA atoms. To solve this problem, we directly add Gaussian noises to the coordinates instead to generate approximate conformers. Specifically, the correlated conformer $\mathcal{P}_2 = (\mathcal{S}_2, \mathcal{R}_2)$ is defined by $\mathcal{S}_2 = \mathcal{S}_1, \mathcal{R}_2 = \mathcal{R}_1 + \epsilon$, where $\epsilon \in \mathbb{R}^{n_a \times 3}$ is the noise drawn from a Gaussian distribution.

## 4 Related Work

**Pre-training Methods on Proteins.** Self-supervised pre-training methods have been widely used to acquire co-evolutionary information from large-scale protein sequence corpus, inducing performant protein language models (PLMs) [20, 53, 61, 50]. Typical sequence pre-training methods include masked protein modeling [20, 61, 50] and contrastive learning [53]. The pre-trained PLMs have achieved impressive performance on a variety of downstream tasks for structure and function prediction [59, 81]. Recent works have also studied pre-training on unlabeled protein structures for generalizable representations, covering contrastive learning [87, 33], self-prediction of geometric quantities [87, 10] and denoising score matching [29, 76]. Compared with existing works, our methods model the joint distribution of sequences and structures via diffusion models, which captures both co-evolutionary information and detailed structural characteristics.

**Diffusion Probabilistic Models (DPMs).** DPM was first proposed in Sohl-Dickstein et al. [63] and has been recently rekindled for its strong performance on image and waveform generation [37, 11]. While DPMs are commonly used for modeling continuous data, there has also been research exploring discrete DPMs that have achieved remarkable results on generating texts [4, 49], graphs [71] and images [38]. Inspired by these progresses, DPMs have been adopted to solve problems in chemistry and biology domain, including molecule generation [82, 39, 78, 43], molecular representation learning [52], protein structure prediction [77], protein-ligand binding [15], protein design [2, 54, 40, 74] and motif-scaffolding [69]. In alignment with recent research efforts focused on diffusion-based image representation learning [1], this work presents a novel investigation into how DPMs can contribute to protein representation learning.

Now we discuss the relationship between our method and previous works.

**Advantages of joint denoising.** Compared with previous diffusion models focusing on either protein sequences [83] or structures [29] or cross-modal contrastive learning [12, 84], in this work, we

Table 1: Atom-level results on Atom3D tasks.

| | Method | PIP | MSP | RES | PSR | | Mean Rank |
|---|---|---|---|---|---|---|---|
| | | AUROC | AUROC | Accuracy | Global $\rho$ | Mean $\rho$ | |
| | GearNet-Edge | 0.868±0.002 | 0.633±0.067 | 0.441±0.001 | 0.782±0.021 | 0.488 ±0.012 | 7.6 |
| w/ pre-training | Denoising Score Matching | 0.877±0.002 | 0.629±0.040 | 0.448±0.001 | 0.813±0.003 | 0.518±0.020 | 5.2 |
| | Residue Type Prediction | 0.879±0.004 | 0.620±0.027 | 0.449±0.001 | 0.826±0.020 | 0.518±0.018 | 4.4 |
| | Distance Prediction | 0.872±0.001 | 0.677±0.020 | 0.422±0.001 | **0.840±0.020** | 0.522±0.004 | 4.0 |
| | Angle Prediction | 0.878±0.001 | 0.642±0.013 | 0.419±0.001 | 0.813±0.007 | 0.503±0.012 | 6.2 |
| | Dihedral Prediction | 0.878±0.004 | 0.591±0.008 | 0.414±0.001 | 0.821±0.002 | 0.497±0.004 | 6.8 |
| | Multiview Contrast | 0.871±0.003 | 0.646±0.006 | 0.368±0.001 | 0.805±0.005 | 0.502±0.009 | 7.2 |
| | **DiffPreT** | 0.880±0.005 | 0.680±0.018 | 0.452±0.001 | 0.821±0.007 | 0.533±0.006 | 2.4 |
| | **SiamDiff** | **0.884±0.003** | **0.698±0.020** | **0.460±0.001** | 0.829±0.012 | **0.546±0.018** | **1.2** |

Table 2: Residue-level results on EC and Atom3D tasks.

| | Method | EC | | MSP | PSR | | Mean Rank |
|---|---|---|---|---|---|---|---|
| | | AUPR | $F_{max}$ | AUROC | Global $\rho$ | Mean $\rho$ | |
| | GearNet-Edge | 0.837±0.002 | 0.811±0.001 | 0.644±0.023 | 0.763±0.012 | 0.373±0.021 | 7.8 |
| w/ pre-training | Denoising Score Matching | 0.859±0.003 | 0.840±0.001 | 0.645±0.028 | 0.795±0.027 | 0.429±0.017 | 5.0 |
| | Residue Type Prediction | 0.851±0.002 | 0.826±0.005 | 0.636±0.003 | 0.828±0.005 | 0.480±0.031 | 5.4 |
| | Distance Prediction | 0.858±0.003 | 0.836±0.001 | 0.623±0.007 | 0.796±0.017 | 0.416±0.021 | 6.4 |
| | Angle Prediction | 0.873±0.003 | 0.849±0.001 | 0.631±0.041 | 0.802±0.015 | 0.446±0.009 | 4.2 |
| | Dihedral Prediction | 0.858±0.001 | 0.840±0.001 | 0.568±0.022 | 0.732±0.021 | 0.398±0.022 | 7.2 |
| | Multiview Contrast | 0.875±0.003 | **0.857±0.003** | **0.713±0.036** | 0.752±0.012 | 0.388±0.015 | 4.0 |
| | **DiffPreT** | 0.864±0.002 | 0.844±0.001 | 0.673±0.042 | 0.815±0.008 | 0.505±0.007 | 3.2 |
| | **SiamDiff** | **0.878±0.003** | **0.857±0.003** | 0.700±0.043 | **0.856±0.007** | **0.521±0.016** | **1.2** |

perform joint diffusion on both modalities. Note that given a sequence $\mathcal{S}$ and a structure $\mathcal{R}$ that exist in the nature with high probability, the sequence-structure tuple $\mathcal{P} = (\mathcal{S}, \mathcal{R})$ may not be a valid state of this protein. Consequently, instead of modeling the marginal or conditional distribution, we model the joint distribution of protein sequences and structures.

**Connection with diffusion models.** Diffusion models excel in image and text generation [17, 49] and have been applied to unsupervised representation learning [1]. Previous works explored denoising objectives [23, 9] but lacked explicit supervision for different conformers, while our method incorporates mutual prediction between siamese diffusion trajectories to capture conformer correlation and regularize protein structure manifold.

**Difference with denoising distance matching.** While previous works rely on perturbing distance matrices [52, 29], which can violate the triangular inequality and produce negative values, our approach directly adds noise at atom coordinates, as demonstrated in Xu et al. [82]. This distinction allows us to address the limitations associated with denoising distance matching algorithms used in molecule and protein generation and pre-training.

**Comparison with other deep generative models.** Self-supervised learning essentially learns an Energy-Based Model (EBM) for modeling data distribution [48], making VAE [45], GAN [26], and normalizing flow [60] applicable for pre-training. However, these models limit flexibility or fail to acquire high sampling quality and diversity compared to diffusion models [54]. Therefore, we focus on using diffusion models for pre-training and leave other generative models for future work.

## 5 Experiments

### 5.1 Experimental Setups

**Pre-training datasets.** Following Zhang et al. [87], we pre-train our models with the AlphaFold protein structure database v1 [44, 70], including 365K proteome-wide predicted structures.

**Downstream benchmark tasks.** In our evaluation, we assess EC prediction task [24] for catalysis behavior of proteins and four ATOM3D tasks [68]. The EC task involves 538 binary classification problems for Enzyme Commission (EC) numbers. We use dataset splits from Gligorijević et al. [24] with a 95% sequence identity cutoff. The ATOM3D tasks include Protein Interface Prediction (PIP),

Mutation Stability Prediction (MSP), Residue Identity (RES), and Protein Structure Ranking (PSR) with different dataset splits based on sequence identity or competition year. Details are in App. D.

**Baseline methods.** In our evaluation, we utilize GearNet-Edge as the underlying model for both atom- and residue-level structures. GearNet-Edge incorporates various types of edges and edge-type-specific convolutions, along with message passing between edges, to model protein structures effectively. We compare our proposed methods with several previous protein structural pre-training algorithms, including multiview contrastive learning [87], denoising score matching [29], and four self-prediction methods (residue type, distance, angle, and dihedral prediction) [87]. For residue-level tasks, we include EC, MSP, and PSR in our evaluation, while PIP and RES tasks are specifically designed for atom-level models. Besides, we exclude EC from the atom-level evaluation due to the limited presence of side-chain atoms in the downloaded PDB dataset.

**Training and evaluation.** We pre-train our model for 50 epochs on the AlphaFold protein structure database following Zhang et al. [87] and fine-tune it for 50 epochs on EC, MSP, and PSR. However, due to time constraints, we only fine-tune the models for 10 epochs on the RES and PIP datasets. Results are reported as mean and standard deviation across three seeds (0, 1, and 2). Evaluation metrics include $F_{max}$ and AUPR for EC, AUROC for PIP and MSP, Spearman's $\rho$ for PSR, and micro-averaged accuracy for RES. More details about experimental setup can be found in App. D.

## 5.2   Experimental Results

Tables 1 and 2 provide a comprehensive overview of the results obtained by GearNet-Edge on both atom- and residue-level benchmark tasks. The tables clearly demonstrate that both DiffPreT and SiamDiff exhibit significant improvements over GearNet-Edge without pre-training on both levels, underscoring the effectiveness of our pre-training methods.

An interesting observation from the tables is that previous pre-training methods tend to excel in specific tasks while showing limitations in others. For instance, Multiview Contrast, designed for capturing similar functional motifs [87], struggles with structural intricacies and local atomic interactions, resulting in lower performance on tasks like Protein Interface Prediction (PIP), Protein Structure Ranking (PSR), and Residue Identity (RES). Self-prediction methods excel at capturing structural details or residue type dependencies but show limitations in function prediction tasks, such as Enzyme Commission (EC) number prediction and Mutation Stability Prediction (MSP), and do not consistently improve performance on both atom and residue levels.

In contrast, our DiffPreT approach achieves top-3 performance in nearly all considered tasks, showcasing its versatility and effectiveness across different evaluation criteria. Moreover, SiamDiff surpasses all other pre-training methods, achieving the best results in 6 out of 7 tasks, establishing it as the state-of-the-art pre-training approach. These results provide compelling evidence that our joint diffusion pre-training strategy successfully captures the intricate interactions between different proteins (PIP), captures local structural details (RES) and global structural characteristics (PSR), and extracts informative features crucial for accurate function prediction (EC) across various tasks.

## 5.3   Ablation Study

To analyze the effect of different components of SiamDiff, we perform ablation study on atom-level tasks and present results in Table 3. We first examine two degenerate settings of joint diffusion, *i.e.*, "*w/o* sequence diffusion" and "*w/o* structure diffusion". These settings lead to a deterioration in performance across all benchmark

Table 3: Ablation study on atom-level Atom3D tasks.

| Method | PIP | MSP | RES | PSR |
|---|---|---|---|---|
| | AUROC | AUROC | Accuracy | Global $\rho$ |
| GearNet-Edge | 0.868±0.002 | 0.633±0.067 | 0.441±0.001 | 0.782±0.021 |
| **SiamDiff** | **0.884±0.003** | **0.698±0.020** | **0.460±0.001** | **0.829±0.008** |
| *w/o seq. diff.* | 0.873±0.004 | 0.695±0.002 | 0.443±0.001 | 0.803±0.010 |
| *w/o struct. diff.* | 0.878±0.003 | 0.652±0.021 | 0.456±0.001 | 0.805±0.005 |
| *w/o MI max.* | 0.880±0.005 | 0.680±0.018 | 0.452±0.001 | 0.821±0.007 |
| *w/ small noise* | 0.875±0.002 | 0.646±0.031 | 0.444±0.001 | 0.828±0.005 |
| *w/ large noise* | 0.867±0.003 | 0.683±0.020 | 0.443±0.001 | 0.819±0.011 |

tasks, highlighting the importance of both sequence diffusion for residue type identification in RES and structure diffusion for capturing the structural stability of mutation effects in MSP. Next, we compare SiamDiff with DiffPreT, which lacks mutual information maximization between correlated conformers. The consistent improvements observed across all tasks indicate the robustness and effectiveness of our proposed mutual information maximization scheme.

Besides, we compare our method to baselines with fixed small ($T = 1$) and large ($T = 100$) noise levels to demonstrate the benefits of multi-scale denoising in diffusion pre-training. Interestingly, we observe that denoising with large noise enhances performance on MSP by capturing significant structural changes that lead to structural instability, while denoising with small noise improves performance in PSR by capturing fine-grained details for protein structure assessment. By incorporating multi-scale noise, we eliminate the need for manual tuning of the noise level as a hyperparameter and leverage the advantages of both large- and small-scale noise, as evidenced in the table.

### 5.4 Combine with Protein Language Models

Protein language models (PLMs) have recently become a standard method for extracting representations from protein sequences, such as ESM [50]. However, these methods are unable to directly handle structure-related tasks

Table 4: ESM2-650M-GearNet on residue-level tasks.

| Method | EC | | MSP | PSR | |
|---|---|---|---|---|---|
| | AUPR | $F_{max}$ | AUROC | Global $\rho$ | Mean $\rho$ |
| GearNet-Edge | 0.837±0.002 | 0.811±0.001 | 0.664±0.023 | 0.764±0.012 | 0.373±0.021 |
| w/ SiamDiff | 0.878±0.003 | 0.857±0.003 | **0.700±0.043** | 0.856±0.007 | 0.521±0.016 |
| ESM-GearNet | 0.904±0.002 | 0.890±0.002 | 0.685±0.027 | 0.829±0.013 | 0.595±0.010 |
| w/ SiamDiff | **0.907±0.001** | **0.897±0.001** | 0.692±0.010 | **0.863±0.009** | **0.656±0.011** |

in Atom3D without using protein structures as input. A recent solution addresses this by feeding residue representations outputted by ESM into the protein structure encoder GearNet [86]. To showcase the potential of SiamDiff on PLM-based encoders, we pre-trained the ESM-GearNet encoder using SiamDiff and evaluated its performance on residue-level tasks. Considering the model capacity and computational budget, we selected ESM-2-650M as the base PLM. The results in Table 4 demonstrate the performance improvements obtained by introducing the PLM component in ESM-GearNet. Furthermore, after pre-training with SiamDiff, ESM-GearNet achieves even better performance on all tasks, especially on PSR where ESM-only representations are not indicative for structure ranking. This highlights the benefits of our method for PLM-based encoders.

In addition, we provide experimental results about pre-training datasets in App. E, multimodal baselines in App. F, different diffusion strategies in App. G, and different backbone models in App. H.

## 6 Conclusions

In this work, we propose the DiffPreT approach to pre-train a protein encoder by sequence-structure joint diffusion modeling, which captures the inter-atomic interactions within structure and the residue type dependencies along sequence. We further propose the SiamDiff method to enhance DiffPreT by additionally modeling the correlation between different conformers of one protein. Extensive experiments on diverse types of tasks and on both atom- and residue-level structures verify the competitive performance of DiffPreT and the superior performance of SiamDiff.

## Acknowledgments

The authors would like to thank Zhaocheng Zhu, Chence Shi, Jiarui Lu, Huiyu Cai, Xinyu Yuan and Bozitao Zhong for their helpful discussions and comments.

This project is supported by AIHN IBM-MILA partnership program, the Natural Sciences and Engineering Research Council (NSERC) Discovery Grant, the Canada CIFAR AI Chair Program, collaboration grants between Microsoft Research and Mila, Samsung Electronics Co., Ltd., Amazon Faculty Research Award, Tencent AI Lab Rhino-Bird Gift Fund, a NRC Collaborative R&D Project (AI4D-CORE-06) as well as the IVADO Fundamental Research Project grant PRF-2019-3583139727.

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

# A  More Related Works

**Protein Structure Encoder.** The community witnessed a surge of research interests in learning informative protein structure representations using structure-based encoders. The encoders are designed to capture protein structural information on different granularity, including residue-level structures [24, 87, 80], atom-level structures [34, 41, 73] and protein surfaces [22, 66, 64]. In this work, we focus on pre-training a typical residue-level structure encoder, *i.e.*, GearNet-Edge [87], and a typical atom-level structure encoder, *i.e.*, GVP [41].

**Mutual Information (MI) Estimation and Maximization.** MI can measure both the linear and non-linear dependency between random variables. Some previous works [7, 35] try to use neural networks to estimate the lower bound of MI, including Donsker-Varadhan representation [19], Jensen-Shannon divergence [21] and Noise-Contrastive Estimation (NCE) [30, 31]. The optimization with InfoNCE loss [57] maximizes a lower bound of MI and is broadly shown to be a superior representation learning strategy [13, 32, 79, 51, 87]. In this work, we adopt the MI lower bound proposed by Liu et al. [52] with two conditional log-likelihoods, and we formulate the learning objective by mutually denoising the multimodal diffusion processes of two correlated proteins.

## A.1  Broader Impacts and Limitations

The main objective of this research project is to enhance protein representations by utilizing joint pre-training using a vast collection of unlabeled protein structures. Unlike traditional unimodel pre-training methods, our approach takes advantage of both sequential and structural information, resulting in superior representations. This advantage allows for more comprehensive analysis of protein research and holds potential benefits for various real-world applications, including protein function prediction and sequence design.

**Limitations.** In this paper, we limit our pre-training dataset to less than 1M protein structures. However, considering the vast coverage of the AlphaFold Protein Structure Database, which includes over 200 million proteins, it becomes feasible to train more advanced and extensive protein encoders on larger datasets in the future. Furthermore, an avenue for future exploration is the incorporation of conformer-related information during pre-training and the development of improved noise schedules for multi-scale denoising pre-training. It is important to acknowledge that powerful pretrained models can potentially be misused for harmful purposes, such as the design of dangerous drugs. We anticipate that future studies will address and mitigate these concerns.

# B  Details of SiamDiff

In this section, we discuss details of our SiamDiff method. We first describe the parameterization of the generation process $p_\theta(\mathcal{P}_1^{t-1}|\mathcal{P}_1^t, \boldsymbol{\mathcal{P}}_2^t)$ in Sec. B.1, derive the pre-training objective in Sec. B.2, and discuss some modifications when applied on residue-level models in Sec. 3.4.

## B.1  Parameterization of Generation Process

Remember that we use $\boldsymbol{\mathcal{P}}_1^{0:T}$ and $\boldsymbol{\mathcal{P}}_2^{0:T}$ to denote the representation of the siamese trajectories $\mathcal{P}_1^{0:T}$ and $\mathcal{P}_2^{0:T}$, respectively. Different from the generation process in traditional diffusion models, the parameterization of $p_\theta(\mathcal{P}_1^{t-1}|\mathcal{P}_1^t, \boldsymbol{\mathcal{P}}_2^t)$ should inject information from $\mathcal{P}_2^t$. Therefore, we use the extracted residue and atom representations (denoted as $\boldsymbol{a}_2^t$ and $\boldsymbol{h}_2^t$) of $\mathcal{P}_2^t$ for this denoising step. Given the conditional independence in (2), this generation process can be decomposed into that on protein structures and sequences similarly in Sec. 2.3.

---

**Algorithm 1** SiamDiff Pre-Training

---

**Input:** training dataset $\mathcal{D}$, learning rate $\alpha$
**Output:** trained encoder $\phi_\theta$

1: **for** $(\mathcal{S}, \mathcal{R})$ in $\mathcal{D}$ **do**
2:     $(\mathcal{S}_1, \mathcal{R}_1) \leftarrow (\mathcal{S}, \mathcal{R})$;
3:     $(\mathcal{S}_2, \mathcal{R}_2) \leftarrow (\mathcal{S}, \text{perturb}(\mathcal{R}, \epsilon))$;
4:     sample noise scale $t \in \{1...T\}$;
5:     $\mathcal{R}_1^t \sim \mathcal{N}(\sqrt{\bar{\alpha}_t}\mathcal{R}_1, (1 - \bar{\alpha}_t)I)$;
6:     $\mathcal{R}_2^t \sim \mathcal{N}(\sqrt{\bar{\alpha}_t}\mathcal{R}_2, (1 - \bar{\alpha}_t)I)$;
7:     $\mathcal{S}_1^t, \mathcal{S}_2^t \sim \text{random\_mask}(\mathcal{S}, t)$
8:     $\boldsymbol{h}_1 \leftarrow \phi_\theta(\mathcal{S}_1^t, \mathcal{R}_1^t)$;
9:     $\boldsymbol{h}_2 \leftarrow \phi_\theta(\mathcal{S}_2^t, \mathcal{R}_2^t)$;
10:    $\mathcal{L}^{(\mathcal{R})} = \mathcal{L}(\mathcal{R}_1, \boldsymbol{h}_2) + \mathcal{L}(\mathcal{R}_2, \boldsymbol{h}_1)$;
11:    $\mathcal{L}^{(\mathcal{S})} = \mathcal{L}(\mathcal{S}_1, \boldsymbol{h}_2) + \mathcal{L}(\mathcal{S}_2, \boldsymbol{h}_1)$;
12:    $\theta \leftarrow \theta - \alpha\nabla_\theta(\mathcal{L}^{(\mathcal{R})} + \mathcal{L}^{(\mathcal{S})})$;
13: **end for**

---

**Generation process on protein structures.** As in (4), modeling the generation process of protein structures is to model the noise on $\mathcal{R}_1^t$ and gradually decorrupt the noisy structure. This can be parameterized with a noise prediction network $\epsilon_\theta(\mathcal{P}_1^t, \boldsymbol{\mathcal{P}}_2^t, t)$ that is translation-invariant and rotation-equivariant *w.r.t.* $\mathcal{R}_1^t$. Besides, the noise applied on $\mathcal{R}_1^t$ should not change with transformations on $\mathcal{R}_2^t$, so $\epsilon_\theta$ should be SE(3)-invariant *w.r.t.* $\mathcal{R}_2^t$.

To achieve these goals, we build our noise prediction network with atom representations $\boldsymbol{a}_2^t$ (which is SE(3)-invariant *w.r.t.* $\mathcal{R}_2^t$) and atom coordinates $\boldsymbol{r}_1^t$ (which is SE(3)-equivariant *w.r.t.* $\mathcal{R}_1^t$). We define an equivariant output similarly as in DiffPreT. Specifically, we have

$$[\epsilon_\theta(\mathcal{P}_1^t, \boldsymbol{\mathcal{P}}_2^t, t)]_i = \sum_{j \in \mathcal{N}_1^t(i)} m_{i,j} \cdot \frac{\boldsymbol{r}_{1i}^t - \boldsymbol{r}_{1j}^t}{\|\boldsymbol{r}_{1i}^t - \boldsymbol{r}_{1j}^t\|_2}, \text{ with } m_{i,j} = \text{MLP}(\boldsymbol{a}_{2i}^t, \boldsymbol{a}_{2j}^t, \text{MLP}(\|\boldsymbol{r}_{1i}^t - \boldsymbol{r}_{1j}^t\|_2)),$$

where $\mathcal{N}_1^t(i)$ denotes the neighbors of the atom $i$ in the corresponding graph of $\mathcal{P}_1^t$. Note that $\epsilon_\theta(\mathcal{P}_1^t, \boldsymbol{\mathcal{P}}_2^t, t)$ achieves the equivariance requirement, as $m_{i,j}$ is SE(3)-invariant *w.r.t.* $\mathcal{R}_1^t$ and $\mathcal{R}_2^t$ while $\boldsymbol{r}_{1i}^t - \boldsymbol{r}_{1j}^t$ is translation-invariant and rotation-equivariant *w.r.t.* $\mathcal{R}_1^t$.

**Generation process on protein sequences.** As in (7), the generation process on sequences aims to predict masked residue types in $\mathcal{S}_1^0$ with a predictor $\tilde{p}_\theta$. In our setting of mutual prediction, we define the predictor based on representations of the same residues in $\mathcal{S}_2^t$, which are also masked. Hence, for each masked residue $i$ in $\mathcal{S}_2^t$, we feed its representation $\boldsymbol{h}_{2i}^t$ to an MLP and predict the type of the corresponding residue type $s_{1i}^0$ in $\mathcal{S}_1^0$:

$$\tilde{p}_\theta(\mathcal{S}_1^0|\mathcal{P}_1^t, \boldsymbol{\mathcal{P}}_2^t) = \prod_i \tilde{p}_\theta(s_{1i}^0|\mathcal{P}_1^t, \boldsymbol{\mathcal{P}}_2^t) = \prod_i \text{Softmax}(s_{1i}^0|\text{MLP}(\boldsymbol{h}_{2i}^t)),$$

where the softmax function is applied over all residue types.

### B.2 Pre-Training Objective

Given the defined forward and reverse process on two trajectories, we now derive the pre-training objective based on the mutual diffusion loss in (13). We take the term $\mathcal{L}^{(2 \to 1)}$ for example and its counterpart can be derived in the same way. The objective can be decomposed into a structure loss $\mathcal{L}^{(\mathcal{R}, 2 \to 1)}$ and a sequence loss $\mathcal{L}^{(\mathcal{S}, 2 \to 1)}$:

$$\mathcal{L}^{(\mathcal{R}, 2 \to 1)} := \mathbb{E}\left[\sum_{t=1}^T D_{\text{KL}}\left(q(\mathcal{R}_1^{t-1}|\mathcal{R}_1^t, \mathcal{R}_1^0)||p_\theta(\mathcal{R}_1^{t-1}|\mathcal{P}_1^t, \boldsymbol{\mathcal{P}}_2^t))\right)\right], \tag{15}$$

$$\mathcal{L}^{(\mathcal{S}, 2 \to 1)} := \mathbb{E}\left[\sum_{t=1}^T D_{\text{KL}}\left(q(\mathcal{S}_1^{t-1}|\mathcal{S}_1^t, \mathcal{S}_1^0)||p_\theta(\mathcal{S}_1^{t-1}|\mathcal{P}_1^t, \boldsymbol{\mathcal{P}}_2^t))\right)\right]. \tag{16}$$

Based on the derivation in Sec. 2.3, the structure loss $\mathcal{L}^{(\mathcal{R}, 2 \to 1)}$ can be simplified as

$$\mathcal{L}_{\text{simple}}^{(\mathcal{R}, 2 \to 1)} = \sum_{t=1}^T \mathbb{E}_{\epsilon \sim \mathcal{N}(0, I)}\left[\|\epsilon - \epsilon_\theta(\mathcal{P}_1^t, \boldsymbol{\mathcal{P}}_2^t, t)\|_2^2\right], \tag{17}$$

and the sequence loss $\mathcal{L}^{(\mathcal{S}, 2 \to 1)}$ can be simplified as

$$\mathcal{L}_{\text{simple}}^{(\mathcal{S}, 2 \to 1)} = \sum_{t=1}^T \sum_i \text{CE}\left(s_{1i}^0, \tilde{p}_\theta(s_{1i}^0|\mathcal{P}_1^t, \boldsymbol{\mathcal{P}}_2^t)\right). \tag{18}$$

Then, the final objective in Sec. 3.3 can be easily derived.

## C    Proofs

In this section, we provide proofs for propositions in Sec. 2 and Sec. 3. Due to the similarity between the two methods, all propositions are restated for SiamDiff. DiffPreT can be seen as a special case that two siamese trajectories collapse into one.

### C.1    Proof of Proposition 1

For notations, we use the bold symbol to denote the representation of an object and use $\boldsymbol{P}_1^{0:T}$ and $\boldsymbol{P}_2^{0:T}$ to denote the corresponding random variables of representations of the siamese trajectories $\boldsymbol{\mathcal{P}}_1^{0:T}$ and $\boldsymbol{\mathcal{P}}_2^{0:T}$.

**Proposition 1** *With some approximations, the mutual information between representations of two siamese trajectories is lower bounded by:*

$$I(\boldsymbol{P}_1^{0:T}; \boldsymbol{P}_2^{0:T}) \geq -\frac{1}{2}(\mathcal{L}^{(2\to1)} + \mathcal{L}^{(1\to2)}) + C,$$

*where $C$ is a constant independent of our encoder and the term from trajectory $\mathcal{P}_b^{0:T}$ to $\mathcal{P}_a^{0:T}$ is defined as*

$$\mathcal{L}^{(b\to a)} := \mathbb{E}_{\mathcal{P}_a^{0:T}, \mathcal{P}_b^{0:T}} \left[ \sum_{t=1}^{T} D_{KL} \left( q(\mathcal{P}_a^{t-1}|\mathcal{P}_a^t, \mathcal{P}_a^0) || p(\mathcal{P}_a^{t-1}|\mathcal{P}_a^t, \boldsymbol{P}_b^{0:T})) \right) \right],$$

*with $b \to a$ being either $2 \to 1$ or $1 \to 2$.*

**Proof.**    First, the mutual information between representations of two trajectories is defined as:

$$I(\boldsymbol{P}_1^{0:T}; \boldsymbol{P}_2^{0:T}) = \mathbb{E}_{\boldsymbol{\mathcal{P}}_1^{0:T}, \boldsymbol{\mathcal{P}}_2^{0:T} \sim p(\boldsymbol{P}_1^{0:T}, \boldsymbol{P}_2^{0:T})} \left[ \log \frac{p(\boldsymbol{\mathcal{P}}_1^{0:T}, \boldsymbol{\mathcal{P}}_2^{0:T})}{p(\boldsymbol{\mathcal{P}}_1^{0:T}) p(\boldsymbol{\mathcal{P}}_2^{0:T})} \right], \tag{19}$$

where the joint distribution is defined as $p(\boldsymbol{P}_1^{0:T}, \boldsymbol{P}_2^{0:T}) = p(\boldsymbol{P}_1^0, \boldsymbol{P}_2^0) q(\boldsymbol{P}_1^{1:T}|\boldsymbol{P}_1^0) q(\boldsymbol{P}_2^{1:T}|\boldsymbol{P}_2^0)$. Next, we can derive a lower bound with this definition:

$$I(\boldsymbol{P}_1^{0:T}; \boldsymbol{P}_2^{0:T}) = \mathbb{E} \left[ \log \frac{p(\boldsymbol{\mathcal{P}}_1^{0:T}, \boldsymbol{\mathcal{P}}_2^{0:T})}{p(\boldsymbol{\mathcal{P}}_1^0) q(\boldsymbol{\mathcal{P}}_1^{1:T}|\boldsymbol{\mathcal{P}}_1^0) p(\boldsymbol{\mathcal{P}}_2^0) q(\boldsymbol{\mathcal{P}}_2^{1:T}|\boldsymbol{\mathcal{P}}_2^0)} \right]$$

$$\geq \mathbb{E} \left[ \log \frac{p(\boldsymbol{\mathcal{P}}_1^{0:T}, \boldsymbol{\mathcal{P}}_2^{0:T})}{\sqrt{p(\boldsymbol{\mathcal{P}}_1^0) p(\boldsymbol{\mathcal{P}}_2^0) q(\boldsymbol{\mathcal{P}}_1^{1:T}|\boldsymbol{\mathcal{P}}_1^0) q(\boldsymbol{\mathcal{P}}_2^{1:T}|\boldsymbol{\mathcal{P}}_2^0)}} \right]$$

$$= \frac{1}{2} \mathbb{E} \left[ \log \frac{p(\boldsymbol{\mathcal{P}}_1^{0:T}, \boldsymbol{\mathcal{P}}_2^{0:T})^2}{p(\boldsymbol{\mathcal{P}}_1^0) p(\boldsymbol{\mathcal{P}}_2^0) q(\boldsymbol{\mathcal{P}}_1^{1:T}|\boldsymbol{\mathcal{P}}_1^0)^2 q(\boldsymbol{\mathcal{P}}_2^{1:T}|\boldsymbol{\mathcal{P}}_2^0)^2} \right]$$

$$= \frac{1}{2} \mathbb{E} \left[ \log \frac{p(\boldsymbol{\mathcal{P}}_1^{0:T}, \boldsymbol{\mathcal{P}}_2^{0:T})}{p(\boldsymbol{\mathcal{P}}_2^0) q(\boldsymbol{\mathcal{P}}_1^{1:T}|\boldsymbol{\mathcal{P}}_1^0) q(\boldsymbol{\mathcal{P}}_2^{1:T}|\boldsymbol{\mathcal{P}}_2^0)} + \log \frac{p(\boldsymbol{\mathcal{P}}_1^{0:T}, \boldsymbol{\mathcal{P}}_2^{0:T})}{p(\boldsymbol{\mathcal{P}}_1^0) q(\boldsymbol{\mathcal{P}}_1^{1:T}|\boldsymbol{\mathcal{P}}_1^0) q(\boldsymbol{\mathcal{P}}_2^{1:T}|\boldsymbol{\mathcal{P}}_2^0)} \right]$$

$$= \frac{1}{2} \mathbb{E} \left[ \log \frac{p(\boldsymbol{\mathcal{P}}_1^{0:T}|\boldsymbol{\mathcal{P}}_2^{0:T})}{q(\boldsymbol{\mathcal{P}}_1^{1:T}|\boldsymbol{\mathcal{P}}_1^0)} + \log \frac{p(\boldsymbol{\mathcal{P}}_2^{0:T}|\boldsymbol{\mathcal{P}}_1^{0:T})}{q(\boldsymbol{\mathcal{P}}_2^{1:T}|\boldsymbol{\mathcal{P}}_2^0)} \right].$$

However, since the distribution of representations are intractable to sample for optimization, we instead sample the trajectories $\mathcal{P}_1^{0:T}$ and $\mathcal{P}_2^{0:T}$ from our defined diffusion process, *i.e.*, $p(\mathcal{P}_1^{0:T}, \mathcal{P}_2^{0:T}) = p(\mathcal{P}_1^0, \mathcal{P}_2^0) q(\mathcal{P}_1^{1:T}|\mathcal{P}_1^0) q(\mathcal{P}_2^{1:T}|\mathcal{P}_2^0)$. Besides, instead of predicting representations, we use the representations from one trajectory to recover the other trajectory, which reflects more information than its representation. With these approximations, the lower bound above can be further written as:

$$\frac{1}{2} \mathbb{E} \left[ \log \frac{p(\boldsymbol{\mathcal{P}}_1^{0:T}|\boldsymbol{\mathcal{P}}_2^{0:T})}{q(\boldsymbol{\mathcal{P}}_1^{1:T}|\boldsymbol{\mathcal{P}}_1^0)} + \log \frac{p(\boldsymbol{\mathcal{P}}_2^{0:T}|\boldsymbol{\mathcal{P}}_1^{0:T})}{q(\boldsymbol{\mathcal{P}}_2^{1:T}|\boldsymbol{\mathcal{P}}_2^0)} \right] \approx \frac{1}{2} \mathbb{E}_{\mathcal{P}_1^{0:T}, \mathcal{P}_2^{0:T}} \left[ \log \frac{p(\mathcal{P}_1^{0:T}|\boldsymbol{\mathcal{P}}_2^{0:T})}{q(\mathcal{P}_1^{1:T}|\mathcal{P}_1^0)} + \log \frac{p(\mathcal{P}_2^{0:T}|\boldsymbol{\mathcal{P}}_1^{0:T})}{q(\mathcal{P}_2^{1:T}|\mathcal{P}_2^0)} \right]$$

We now show the first term on the right hand side can be written as the loss defined in Proposition 1. The derivation is very similar with the proof of Proposition 3 in Xu et al. [82]. We include it here for

completeness:

$$
\begin{aligned}
&\mathbb{E}_{\mathcal{P}_1^{0:T},\mathcal{P}_2^{0:T}}\left[\log\frac{p(\mathcal{P}_1^{0:T}|\boldsymbol{P}_2^{0:T})}{q(\mathcal{P}_1^{1:T}|\mathcal{P}_1^0)}\right]\\
=&\mathbb{E}_{\mathcal{P}_1^{0:T},\mathcal{P}_2^{0:T}}\left[\sum_{t=1}^{T}\log\frac{p(\mathcal{P}_1^{t-1}|\mathcal{P}_1^t,\boldsymbol{P}_2^{0:T})}{q(\mathcal{P}_1^t|\mathcal{P}_1^{t-1})}\right]\\
=&\mathbb{E}_{\mathcal{P}_1^{0:T},\mathcal{P}_2^{0:T}}\left[\log\frac{(\mathcal{P}_1^0|\mathcal{P}_1^1,\boldsymbol{P}_2^{0:T})}{q(\mathcal{P}_1^1|\mathcal{P}_1^0)}+\sum_{t=2}^{T}\log\left(\frac{p(\mathcal{P}_1^{t-1}|\mathcal{P}_1^t,\boldsymbol{P}_2^{0:T})}{q(\mathcal{P}_1^{t-1}|\mathcal{P}_1^t,\mathcal{P}_1^0)}\cdot\frac{q(\mathcal{P}_1^{t-1}|\mathcal{P}_1^0)}{q(\mathcal{P}_1^t|\mathcal{P}_1^0)}\right)\right]\\
=&\mathbb{E}_{\mathcal{P}_1^{0:T},\mathcal{P}_2^{0:T}}\left[-\log q(\mathcal{P}_1^T|\mathcal{P}_1^0)+\log p(\mathcal{P}_1^0|\mathcal{P}_1^1,\boldsymbol{P}_2^{0:T})+\sum_{t=2}^{T}\log\frac{p(\mathcal{P}_1^{t-1}|\mathcal{P}_1^t,\boldsymbol{P}_2^{0:T})}{q(\mathcal{P}_1^{t-1}|\mathcal{P}_1^t,\mathcal{P}_1^0)}\right]\\
=&-\mathbb{E}_{\mathcal{P}_1^{0:T},\mathcal{P}_2^{0:T}}\left[\sum_{t=1}^{T}D_{\mathrm{KL}}\left(q(\mathcal{P}_1^{t-1}|\mathcal{P}_1^t,\mathcal{P}_1^0)||p(\mathcal{P}_1^{t-1}|\mathcal{P}_1^t,\boldsymbol{P}_2^{0:T})\right)\right]+C^{(2\to1)}\\
=&-\mathcal{L}^{(2\to1)}+C^{(2\to1)},
\end{aligned}
$$

where we merge the term $p(\mathcal{P}_1^0|\mathcal{P}_1^1,\boldsymbol{P}_2^{0:T})$ into the sum of KL divergences for brevity and use $C^{(2\to1)}$ to denote the constant independent of our encoder. Note that the counterpart can be derived in the same way. Adding these two terms together finishes the proof of Proposition 1. $\square$

## C.2 Proof of Pre-Training Loss Decomposition

We restate the proposition of pre-training loss decomposition rigorously as below.

**Proposition 2** *Given the assumptions 1) the separation of the diffusion process on protein structures and sequences*

$$
q(\mathcal{P}_a^t|\mathcal{P}_a^{t-1}) = q(\mathcal{R}_a^t|\mathcal{R}_a^{t-1})\cdot q(\mathcal{S}_a^t|\mathcal{S}_a^{t-1}), \tag{20}
$$

*and 2) the conditional independence of the generation process*

$$
p_\theta(\mathcal{P}_a^{t-1}|\mathcal{P}_a^t,\boldsymbol{P}_b^t) = p_\theta(\mathcal{R}_a^{t-1}|\mathcal{P}_a^t,\boldsymbol{P}_b^t)\cdot p_\theta(\mathcal{S}_a^{t-1}|\mathcal{P}_a^t,\boldsymbol{P}_b^t), \tag{21}
$$

*it can be proved that*

$$
\mathcal{L}^{(b\to a)} = \mathcal{L}^{(\mathcal{R},b\to a)} + \mathcal{L}^{(\mathcal{S},b\to a)}, \tag{22}
$$

*where the three loss terms are defined as*

$$
\begin{aligned}
\mathcal{L}^{(b\to a)} &:= \mathbb{E}\left[\sum_{t=1}^{T}D_{KL}\left(q(\mathcal{P}_a^{t-1}|\mathcal{P}_a^t,\mathcal{P}_a^0)||p_\theta(\mathcal{P}_a^{t-1}|\mathcal{P}_a^t,\boldsymbol{P}_b^t))\right)\right],\\
\mathcal{L}^{(\mathcal{R},b\to a)} &:= \mathbb{E}\left[\sum_{t=1}^{T}D_{KL}\left(q(\mathcal{R}_a^{t-1}|\mathcal{R}_a^t,\mathcal{R}_a^0)||p_\theta(\mathcal{R}_a^{t-1}|\mathcal{P}_a^t,\boldsymbol{P}_b^t))\right)\right],\\
\mathcal{L}^{(\mathcal{S},b\to a)} &:= \mathbb{E}\left[\sum_{t=1}^{T}D_{KL}\left(q(\mathcal{S}_a^{t-1}|\mathcal{S}_a^t,\mathcal{S}_a^0)||p_\theta(\mathcal{S}_a^{t-1}|\mathcal{P}_a^t,\boldsymbol{P}_b^t))\right)\right],
\end{aligned}
$$

*with $b\to a$ referring to the term from trajectory $\mathcal{P}_b^{0:T}$ to $\mathcal{P}_a^{0:T}$.*

**Proof.** Let $\mathcal{L}_t^{(\cdot)}$ to denote the t-th KL divergence term in $\mathcal{L}^{(\cdot)}$. Then, we have

$$
\begin{aligned}
\mathcal{L}_t^{(b\to a)} &= D_{\mathrm{KL}}\left(q(\mathcal{P}_a^{t-1}|\mathcal{P}_a^t,\mathcal{P}_a^0)||p_\theta(\mathcal{P}_a^{t-1}|\mathcal{P}_a^t,\boldsymbol{P}_b^{0:T})\right)\\
&= D_{\mathrm{KL}}\left(\left[q(\mathcal{R}_a^{t-1}|\mathcal{R}_a^t,\mathcal{R}_a^0)q(\mathcal{S}_a^{t-1}|\mathcal{S}_a^t,\mathcal{S}_a^0)\right]||\left[p_\theta(\mathcal{R}_a^{t-1}|\mathcal{P}_a^t,\boldsymbol{P}_b^{0:T})p_\theta(\mathcal{S}_a^{t-1}|\mathcal{P}_a^t,\boldsymbol{P}_b^{0:T})\right]\right)\\
&= D_{\mathrm{KL}}\left(q(\mathcal{R}_a^{t-1}|\mathcal{R}_a^t,\mathcal{R}_a^0)||p_\theta(\mathcal{R}_a^{t-1}|\mathcal{P}_a^t,\boldsymbol{P}_b^{0:T})\right) + D_{\mathrm{KL}}\left(q(\mathcal{S}_a^{t-1}|\mathcal{S}_a^t,\mathcal{S}_a^0)||p_\theta(\mathcal{S}_a^{t-1}|\mathcal{P}_a^t,\boldsymbol{P}_b^{0:T})\right)\\
&= \mathcal{L}_t^{(\mathcal{R},b\to a)} + \mathcal{L}_t^{(\mathcal{S},b\to a)},
\end{aligned}
$$

where we use the assumptions (20) and (21) in the second equality. The third equality is due to the additive property of the KL divergence for independent distributions. Adding $T$ KL divergence terms together will lead to (22). $\square$

## C.3  Proof of Simplified Structure Loss

For completeness, we show how to derive the simplified structure loss. The proof is directly adapted from [82].

**Proposition 3** *Given the definition of the forward process*

$$q(\mathcal{R}_a^t|\mathcal{R}_a^{t-1}) = \mathcal{N}(\mathcal{R}_a^t; \sqrt{1-\beta_t}\mathcal{R}_a^{t-1}, \beta_t I), \tag{23}$$

*and the reverse process*

$$p_\theta(\mathcal{R}_a^{t-1}|\mathcal{P}_a^t, \boldsymbol{P}_b^t) = \mathcal{N}(\mathcal{R}_a^{t-1}; \mu_\theta(\mathcal{P}_a^t, \boldsymbol{P}_b^t, t), \sigma_t^2 I), \tag{24}$$

$$\mu_\theta(\mathcal{P}_a^t, \boldsymbol{P}_b^t, t) = \frac{1}{\sqrt{\alpha_t}}\left(\mathcal{R}_a^t - \frac{\beta_t}{\sqrt{1-\bar{\alpha}_t}}\epsilon_\theta(\mathcal{P}_a^t, \boldsymbol{P}_b^t, t)\right), \tag{25}$$

*the structure loss function*

$$\mathcal{L}^{(\mathcal{R}, b\to a)} := \mathbb{E}\left[\sum_{t=1}^T D_{KL}\left(q(\mathcal{R}_a^{t-1}|\mathcal{R}_a^t, \mathcal{R}_a^0)||p_\theta(\mathcal{R}_a^{t-1}|\mathcal{P}_a^t, \boldsymbol{P}_b^t))\right], \tag{26}$$

*can be simplified to*

$$\mathcal{L}^{(\mathcal{R}, b\to a)} = \sum_{t=1}^T \gamma_t \mathbb{E}_{\epsilon\sim\mathcal{N}(0,I)}\left[\|\epsilon - \epsilon_\theta(\mathcal{P}_a^t, \boldsymbol{P}_b^t, t)\|_2^2\right], \tag{27}$$

*where* $\gamma_t = \frac{\beta_t}{2\alpha_t(1-\bar{\alpha}_{t-1})}$ *with* $\alpha_t = 1 - \beta_t$, $\bar{\alpha}_t = \prod_{s=1}^t \alpha_s$ *and* $b \to a$ *is either* $2 \to 1$ *or* $1 \to 2$.

**Proof.**  First, we prove $q(\mathcal{R}_a^t|\mathcal{R}_a^0) = \mathcal{N}(\mathcal{R}_a^t; \sqrt{\bar{\alpha}_t}\mathcal{R}_a^0, (1-\bar{\alpha}_t)I)$. Let $\epsilon_i$ be the standard Gaussian random variable at time step $i$. Then, we have

$$\begin{aligned}
\mathcal{R}_a^t &= \sqrt{\alpha_t}\mathcal{R}_a^{t-1} + \sqrt{\beta_t}\epsilon_t \\
&= \sqrt{\alpha_{t-1}\alpha_t}\mathcal{R}_a^{t-2} + \sqrt{\alpha_{t-1}\beta_{t-1}}\epsilon_{t-1} + \sqrt{\beta_t}\epsilon_t \\
&= \cdots \\
&= \sqrt{\bar{\alpha}_t}\mathcal{R}_a^0 + \sqrt{\alpha_t\alpha_{t-1}...\alpha_2\beta_1}\epsilon_1 + \cdots + \sqrt{\alpha_{t-1}\beta_{t-1}}\epsilon_{t-1} + \sqrt{\beta_t}\epsilon_t,
\end{aligned}$$

which suggests that the mean of $\mathcal{R}_a^t$ is $\sqrt{\bar{\alpha}_t}\mathcal{R}_a^0$ and the variance matrix is $(\alpha_t\alpha_{t-1}...\alpha_2\beta_1 + \cdots + \alpha_{t-1}\beta_{t-1} + \beta_t)I = (1-\bar{\alpha})I$.

Next, we derive the posterior distribution as:

$$\begin{aligned}
q(\mathcal{R}_a^{t-1}|\mathcal{R}_a^t, \mathcal{R}_a^0) &= \frac{q(\mathcal{R}_a^t|\mathcal{R}_a^{t-1})q(\mathcal{R}_a^{t-1}|\mathcal{R}_a^0)}{q(\mathcal{R}_a^t|\mathcal{R}_a^0)} \\
&= \frac{\mathcal{N}(\mathcal{R}_a^t; \sqrt{\alpha_t}\mathcal{R}_a^{t-1}, \beta_t I) \cdot \mathcal{N}(\mathcal{R}_a^{t-1}; \sqrt{\bar{\alpha}_{t-1}}\mathcal{R}_a^0, (1-\bar{\alpha}_{t-1})I)}{\mathcal{N}(\mathcal{R}_a^t; \sqrt{\bar{\alpha}_t}\mathcal{R}_a^0, (1-\bar{\alpha}_t)I)} \\
&= \mathcal{N}(\mathcal{R}_a^{t-1}; \frac{\sqrt{\bar{\alpha}_{t-1}}\beta_t}{1-\bar{\alpha}_t}\mathcal{R}_a^0 + \frac{\sqrt{\alpha_t}(1-\bar{\alpha}_{t-1})}{1-\bar{\alpha}_t}\mathcal{R}_a^t, \frac{1-\bar{\alpha}_{t-1}}{1-\bar{\alpha}_t}\beta_t I).
\end{aligned}$$

Let $\tilde{\beta}_t = \frac{1-\bar{\alpha}_{t-1}}{1-\bar{\alpha}_t}\beta_t$, then the $t$-th KL divergence term can be written as:

$$D_{\text{KL}}\left(q(\mathcal{R}_a^{t-1}|\mathcal{R}_a^t, \mathcal{R}_a^0)||p_\theta(\mathcal{R}_a^{t-1}|\mathcal{P}_a^t, \boldsymbol{P}_b^t)\right)$$

$$=\frac{1}{2\tilde{\beta}_t}\left\|\frac{\sqrt{\bar{\alpha}_{t-1}}\beta_t}{1-\bar{\alpha}_t}\mathcal{R}_a^0 + \frac{\sqrt{\alpha_t}(1-\bar{\alpha}_{t-1})}{1-\bar{\alpha}_t}\mathcal{R}_a^t - \frac{1}{\sqrt{\alpha_t}}\left(\mathcal{R}_a^t - \frac{\beta_t}{\sqrt{1-\bar{\alpha}_t}}\epsilon_\theta(\mathcal{P}_a^t, \boldsymbol{P}_b^t, t)\right)\right\|^2$$

$$=\frac{1}{2\tilde{\beta}_t}\mathbb{E}_\epsilon\left\|\frac{\sqrt{\bar{\alpha}_{t-1}}\beta_t}{1-\bar{\alpha}_t}\cdot\frac{\mathcal{R}_a^t - \sqrt{1-\bar{\alpha}_t}\epsilon}{\sqrt{\bar{\alpha}_t}} + \frac{\sqrt{\alpha_t}(1-\bar{\alpha}_{t-1})}{1-\bar{\alpha}_t}\mathcal{R}_a^t - \frac{1}{\sqrt{\alpha_t}}\left(\mathcal{R}_a^t - \frac{\beta_t}{\sqrt{1-\bar{\alpha}_t}}\epsilon_\theta(\mathcal{P}_a^t, \boldsymbol{P}_b^t, t)\right)\right\|^2$$

$$=\frac{1}{2\tilde{\beta}_t}\cdot\frac{\beta_t^2}{\alpha_t(1-\bar{\alpha}_t)}\mathbb{E}_\epsilon\left\|\epsilon - \epsilon_\theta(\mathcal{P}_a^t, \boldsymbol{P}_b^t, t)\right\|$$

$$=\gamma_t\mathbb{E}_\epsilon\left[\|\epsilon - \epsilon_\theta(\mathcal{P}_a^t, \boldsymbol{P}_b^t, t)\|_2^2\right],$$

which completes the proof.  $\square$

## C.4 Proof of Simplified Sequence Loss

Now we show the equivalence of optimizing sequence loss $\mathcal{L}^{(\mathcal{S}, b \to a)}$ and the masked residue type prediction problem on $\mathcal{S}_a^0$.

**Proposition 4** *Given the definition of reverse process on protein sequences*

$$p_\theta(\mathcal{S}_a^{t-1}|\mathcal{P}_a^t, \boldsymbol{\mathcal{P}}_b^t) \propto \sum_{\tilde{\mathcal{S}}_a^0} q(\mathcal{S}_a^{t-1}|\mathcal{S}_a^t, \tilde{\mathcal{S}}_a^0) \cdot \tilde{p}_\theta(\tilde{\mathcal{S}}_a^0|\mathcal{P}_a^t, \boldsymbol{\mathcal{P}}_b^t), \tag{28}$$

*the sequence loss $\mathcal{L}^{(\mathcal{S}, b \to a)}$ reaches zero when $\tilde{p}_\theta(\tilde{\mathcal{S}}_a^0|\mathcal{P}_a^t, \boldsymbol{\mathcal{P}}_b^t)$ puts all mass on the ground truth $\mathcal{S}_a^0$.*

**Proof.** The loss function can be written as:

$$\mathcal{L}^{(\mathcal{S}, b \to a)} := \mathbb{E}\left[\sum_{t=1}^T D_{\mathrm{KL}}\left(q(\mathcal{S}_a^{t-1}|\mathcal{S}_a^t, \mathcal{S}_a^0)||p_\theta(\mathcal{S}_a^{t-1}|\mathcal{P}_a^t, \boldsymbol{\mathcal{P}}_b^t)\right)\right]$$

$$= \mathbb{E}\left[\sum_{t=1}^T D_{\mathrm{KL}}\left(q(\mathcal{S}_a^{t-1}|\mathcal{S}_a^t, \mathcal{S}_a^0)\middle\|\frac{\sum_{\tilde{\mathcal{S}}_a^0} q(\mathcal{S}_a^{t-1}|\mathcal{S}_a^t, \tilde{\mathcal{S}}_a^0) \cdot \tilde{p}_\theta(\tilde{\mathcal{S}}_a^0|\mathcal{P}_1^t, \boldsymbol{\mathcal{P}}_2^t)}{Z}\right)\right],$$

where $Z$ is the normalization constant. Hence, when $\tilde{p}_\theta(\tilde{\mathcal{S}}_a^0|\mathcal{P}_a^t, \boldsymbol{\mathcal{P}}_b^t)$ puts all mass on the ground truth $\mathcal{S}_a^0$, the distribution $p_\theta(\mathcal{S}_a^{t-1}|\mathcal{P}_a^t, \boldsymbol{\mathcal{P}}_b^t)$ will be identical with $q(\mathcal{S}_a^{t-1}|\mathcal{S}_a^t, \mathcal{S}_a^0)$, which makes the KL divergence become zero. $\square$

# D Experimental Details

In this section, we introduce the details of our experiments. All these methods are developed based on PyTorch and TorchDrug [88].

**Downstream benchmark tasks.** For downstream evaluation, we adopt the EC prediction task [24] and four ATOM3D tasks [68].

1. **Enzyme Commission (EC) number prediction** task aims to predict EC numbers of proteins which describe their catalysis behavior in biochemical reactions. This task is formalized as 538 binary classification problems. We adopt the dataset splits from Gligorijević et al. [24] and use the test split with 95% sequence identity cutoff following Zhang et al. [87].
2. **Protein Interface Prediction (PIP)** requires the model to predict whether two amino acids from two proteins come into contact when the proteins bind (binary classification). The protein complexes of this benchmark are split with 30% sequence identity cutoff.
3. **Mutation Stability Prediction (MSP)** task seeks to predict whether a mutation will increase the stability of a protein complex or not (binary classification). The benchmark dataset is split upon a 30% sequence identity cutoff among different splits.
4. **Residue Identity (RES)** task studies the structural role of an amino acid under its local environment. A model predicts the type of the center amino acid based on its surrounding atomic structure. The environments in different splits are with different protein topology classes.
5. **Protein Structure Ranking (PSR)** predicts global distance test scores of structure predictions submitted to the Critical Assessment of Structure Prediction (CASP) [47] competition. This dataset is split according to the competition year.

**Graph construction.** For atom graphs, we connect atoms with Euclidean distance lower than a distance threshold. For PSR and MSP tasks, we remove all hydrogen atoms following Jing et al. [42]. For residue graphs, we discard all non-alpha-carbon atoms and add three different types of directed edges: sequential edges, radius edges and K-nearest neighbor edges. For sequential edges, two atoms are connected if their sequential distance is below a threshold and these edges are divided into different types according to these distances. For two kinds of spatial edges, we connect atoms according to Euclidean distance and k-nearest neighbors. We further apply a long range interaction filter that removes edges with low sequential distances. We refer readers to Zhang et al. [87] for more details.

**Atom-level backbone models.** To adapt GearNet-Edge to atom-level structures with moderate computational cost, we construct the atom graph by using only the spatial edge with the radius

$d_{\text{radius}} = 4.5\text{Å}$. We concatenate one-hot features of atom types and residue types as node features and concatenate (1) one-hot features of residue types of end nodes, (2) one-hot features of edge types, (3) one-hot features of sequential distance, (4) spatial distance as edge features. The whole model is composed of 6 message passing layers with 128 hidden dimensions and ReLU activation function. For edge message passing, we employ the discretized angles to determine the edge types on the line graph. The final prediction is performed upon the hidden representation concatenated across all layers.

**Residue-level backbone models.** We directly borrow the best hyperparameters reported in the original paper of GearNet-Edge [87]. We adopt the same configuration of relational graph construction, *i.e.*, the sequential distance threshold $d_{\text{seq}} = 3$, the radius $d_{\text{radius}} = 10.0\text{Å}$, the number of neighbors $k = 10$ and the long range interaction cutoff $d_{\text{long}} = 5$. We use one-hot features of residue types as node features and concatenate (1) features of end nodes, (2) one-hot features of edge types, (3) one-hot features of sequential distance, (4) spatial distance as edge features. Then we use 6 message passing layers with 512 hidden dimensions and ReLU as the activation function. For edge message passing, the edge types on the line graph are determined by the discretized angles. The hidden representations in each layer of GearNet will be concatenated for the final prediction.

**Baseline pre-training methods.** Here we briefly introduce the considered baselines. Multiview Contrast aims to maximize the mutual information between correlated views, which are extracted by randomly chosen augmentation functions to capture protein sub-structures. Residue type, distance, angle and dihedral prediction masks single residues, single edges, edge pairs and edge triplets, respectively, and then predict the corresponding properties. Denoising score matching performs denoising on noised pairwise distance matrices based on the learnt representations.

For all baselines in [87], we adopt the original configurations. For Multiview Contrast, we use subsequence cropping that randomly extracts protein subsequences with no more than 50 residues and space cropping that takes all residues within a 15Å Euclidean ball with a random center residue. Then, either an identity function or a random edge masking function with mask rate equal to 0.15 is applied for constructing views. The temperature $\tau$ in the InfoNCE loss function is set as 0.07. We set the number of sampled items in each protein as 256 for Distance Prediction and as 512 for Angle and Dihedral Prediction. The mask rate for Residue Type Prediction is set as 0.15. When masking a residue on atom graphs, we discard all non-backbone atoms and set the residue features as zero. Since the backbone models and tasks in our paper are quite different with those in Guo et al. [29], we re-implement the method on our codebase. We consider 50 different noise levels log-linearly ranging from 0.01 to 10.0.

In DiffPreT, for structure diffusion, we use a sigmoid schedule for variances $\beta_t$ with the lowest variance $\beta_1 = 1e - 4$ and the highest variance $\beta_T = 0.1$. For sequence diffusion, we simply set the cumulative transition probability to [MASK] over time steps as a linear interpolation between minimum mask rate 0.15 and maximum mask rate 1.0. The number of diffusion steps is set as 100. In SiamDiff, we adopt the same hyperparameters for multimodal diffusion models. We set the variance of torsional perturbation noises as $0.1\pi$ on the atom level and that of Gaussian perturbation noises as 0.3 on the residue level when constructing the correlated conformer.

All other optimization configurations for these pre-training methods are reported in Table 5. All methods are pre-trained on 4 Tesla A100 GPUs and Table 5 reports the batch sizes on each GPU.

**Fine-tuning on downstream tasks.** For all models on all downstream tasks, we apply a three-layer MLP head for prediction, the hidden dimension of which is set to the dimension of model outputs. The number of used gpus and batch sizes for each model are chosen according the memory limit. All residue-level tasks are run on 4 V100 GPUs while all atom-level tasks are run on A100 GPUs.

**Evaluation metrics.** We clarify the definitions of $F_{\text{max}}$ (used in EC), global Spearman's $\rho$ (used in PSR) and mean Spearman's $\rho$ (used in PSR) as below:

- **$F_{\text{max}}$** denotes the protein-centric maximum F-score. It first computes the precision and recall for each protein at a decision threshold $t \in [0, 1]$:

$$\text{precision}_i(t) = \frac{\sum_f \mathbb{1}[f \in P_i(t) \cap T_i]}{\sum_f \mathbb{1}[f \in P_i(t)]}, \quad \text{recall}_i(t) = \frac{\sum_f \mathbb{1}[f \in P_i(t) \cap T_i]}{\sum_f \mathbb{1}[f \in T_i]}, \quad (29)$$

Table 5: Optimization configurations for pre-training methods. Here max length denotes the maximum number of residues kept in each protein and lr stands for learning rate.

| Method | Max length | | Batch size | | Optimizer | lr |
|---|---|---|---|---|---|---|
| | residue | atom | residue | atom | | |
| Residue Type Prediction | 100 | 100 | 96 | 64 | Adam | 1e-3 |
| Distance Prediction | 100 | 100 | 128 | 64 | Adam | 1e-3 |
| Angle Prediction | 100 | 100 | 96 | 64 | Adam | 1e-3 |
| Dihedral Prediction | 100 | 100 | 96 | 64 | Adam | 1e-3 |
| Multiview Contrast | - | - | 96 | 64 | Adam | 1e-3 |
| Denoising Score Matching | 200 | 200 | 12 | 12 | Adam | 1e-4 |
| **DiffPreT** | 150 | 100 | 16 | 64 | Adam | 1e-4 |
| **SiamDiff** | 150 | 100 | 16 | 32 | Adam | 1e-4 |

Table 6: Optimization configurations for downstream evaluations. Here max length denotes the maximum number of residues kept in each protein and lr stands for learning rate.

| Task | # GPUS | | Batch size | | Optimizer | lr |
|---|---|---|---|---|---|---|
| | residue | atom | residue | atom | | |
| EC | 4 | N/A | 2 | N/A | Adam | 1e-4 |
| PIP | N/A | 1 | N/A | 8 | Adam | 1e-4 |
| MSP | 4 | 1 | 1 | 8 | Adam | 1e-4 |
| RES | N/A | 4 | N/A | 64 | Adam | 1e-4 |
| PSR | 4 | 1 | 8 | 8 | Adam | 1e-4 |

where $f$ denotes a functional term in the ontology, $T_i$ is the set of experimentally determined functions for protein $i$, $P_i(t)$ is the set of predicted functions for protein $i$ whose scores are greater or equal to $t$, and $\mathbb{1}[\cdot]$ represents the indicator function. After that, the precision and recall are averaged over all proteins:

$$\text{precision}(t) = \frac{1}{M(t)} \sum_i \text{precision}_i(t), \quad \text{recall}(t) = \frac{1}{N} \sum_i \text{recall}_i(t), \tag{30}$$

where $N$ denotes the total number of proteins, and $M(t)$ denotes the number of proteins which contain at least one prediction above the threshold $t$, $i.e.$, $|P_i(t)| > 0$.

Based on these two metrics, the $F_{\text{max}}$ score is defined as the maximum value of F-measure over all thresholds:

$$F_{\text{max}} = \max_t \left\{ \frac{2 \cdot \text{precision}(t) \cdot \text{recall}(t)}{\text{precision}(t) + \text{recall}(t)} \right\}. \tag{31}$$

- **Global Spearman's $\rho$ for PSR** measures the correlation between the predicted global distance test (GDT_TS) score and the ground truth. It computes the Spearman's $\rho$ between the prediction and the ground truth over all test proteins without considering the different biopolymers that these proteins lie in.

- **Mean Spearman's $\rho$ for PSR** also measures the correlation between GDT_TS predictions and the ground truth. However, it first splits all test proteins into multiple groups based on their corresponding biopolymers, then computes the Spearman's $\rho$ within each group, and finally reports the mean Spearman's $\rho$ over all groups.

## E  Results of Pre-Training on Different Sizes of Datasets

In the main paper, we followed the setting in Zhang et al. [87] and used AlphaFold Database v1 as our pre-training dataset for fair comparison. Here, we investigate the impact of pre-training on different dataset sizes. Since previous work by Zhang et al. [87] showed minimal differences between using experimental or predicted structures, we conduct experiments on the AlphaFold Database in this section and do not use PDB as our pre-training dataset. We utilize the preprocessed clustered

Table 7: Atom-level results of pre-training on different sizes of datasets on Atom3D tasks.

| Pre-training Dataset | Size | PSR | | MSP | PIP | RES |
|---|---|---|---|---|---|---|
| | | Global $\rho$ | Mean $\rho$ | AUROC | AUROC | Acc. |
| AlphaFold DB v1 | 365K | 0.829 | 0.546 | 0.698 | **0.884** | **0.460** |
| Clustered AlphaFold DB | 10K | 0.816 | 0.501 | 0.586 | 0.880 | 0.444 |
| Clustered AlphaFold DB | 50K | 0.797 | 0.498 | 0.648 | 0.879 | 0.450 |
| Clustered AlphaFold DB | 100K | 0.805 | 0.540 | 0.685 | 0.882 | 0.454 |
| Clustered AlphaFold DB | 500K | **0.848** | 0.537 | 0.599 | 0.880 | 0.459 |
| Clustered AlphaFold DB | 2.2M | 0.840 | **0.560** | **0.700** | 0.882 | **0.460** |

AlphaFold Database provided in [6], which includes 2.2M non-singleton clusters with an average of 13.2 proteins per cluster and an average pLDDT of 71.59. For each cluster, we use the representative structure from [6]. To explore the effects of dataset size, we pre-train our encoder using 10K, 50K, 100K, 500K, and 2.2M clusters in the database. The results, shown in Table 7, reveal a general trend of increased performance with larger datasets. However, for certain tasks like MSP, the performance does not consistently improve, possibly due to the limited size and variability of the downstream datasets. Overall, scaling the model to the entire AlphaFold Database holds promise for achieving performance gains.

# F   Results of Multimodal Pre-Training Baselines

Table 8: Atom-level results of multimodal pre-training baselines on Atom3D tasks.

| | Method | PSR | | MSP | PIP | RES |
|---|---|---|---|---|---|---|
| | | Global $\rho$ | Mean $\rho$ | AUROC | AUROC | Acc. |
| | GearNet-Edge | 0.782 | 0.488 | 0.633 | 0.868 | 0.441 |
| w/ pre-training | Residue Type Prediction | 0.826 | 0.518 | 0.620 | 0.879 | 0.449 |
| | Residue Type & Distance Prediction | 0.817 | 0.518 | 0.665 | 0.873 | 0.402 |
| | Residue Type & Angle Prediction | **0.837** | 0.524 | 0.642 | 0.878 | 0.415 |
| | Residue Type & Dihedral Prediction | 0.823 | 0.494 | 0.597 | 0.871 | 0.414 |
| | **DiffPreT** | 0.821 | 0.533 | 0.680 | 0.880 | 0.452 |
| | **SiamDiff** | 0.829 | **0.546** | **0.698** | **0.884** | **0.460** |

In Sec. 5, we include all pre-training baselines from existing works, which solely focus on unimodal pre-training objectives and overlook the joint distribution of sequences and structures. In this section, we introduce three additional pre-training baselines that leverage both sequence and structure information. These baselines combine residue type prediction with distance/angle/dihedral prediction objectives. Following established settings, we pre-train the encoder and present the results in Table 8. While the Residue Type & Angle Prediction method achieves higher performance on PSR, there are still substantial gaps compared to our DiffPreT and SiamDiff across other tasks. Notably, the introduction of geometric property prediction tasks leads to a drop in performance on RES, indicating that a simple combination of pre-training objectives diminishes the benefits of each individual objective. Once again, these findings underscore the effectiveness of our methods in modeling the joint distribution of protein sequences and structures.

# G   Results of Different Diffusion Models for Pre-Training

In Sec. 2, we consider joint diffusion models on protein sequences and structures for pre-training. In this section, we explore the performance of different diffusion models when applied for pre-training. The results are shown in Table 9.

First, we simply run diffusion models on protein sequences and structures for pre-training, both of which achieve improvement compared with the baseline GearNet-Edge. By combining two diffusion models, DiffPreT can achieve better performance on PIP and PSR. This advantage will be further

Table 9: Results of different diffusion models for pre-training on atom-level Atom3D tasks.

| | Method | PIP | RES | PSR | |
|---|---|---|---|---|---|
| | | AUROC | Accuracy | Global $\rho$ | Mean $\rho$ |
| | GearNet-Edge | 0.868 | 0.441 | 0.782 | 0.488 |
| w/ pre-training | **DiffPreT** | 0.880 | 0.452 | 0.821 | 0.533 |
| | **SiamDiff** | **0.884** | **0.460** | **0.829** | **0.546** |
| | Sequence Diffusion | 0.879 | 0.456 | 0.802 | 0.508 |
| | Structure Diffusion | 0.877 | 0.448 | 0.813 | 0.518 |
| | Torsional Diffusion | 0.877 | 0.442 | 0.819 | 0.505 |

increased after using siamese diffusion trajectory prediction as shown in Table 3. It can be observed that sequence diffusion achieves better performance than DiffPreT due to the consistency of objectives between pre-training and RES tasks.

Besides, we also consider diffusion models on torsional angles of protein side chains for pre-training. This method has shown its potential in the protein-ligand docking task [15]. For each residue, we randomly select one side-chain torsional angle and add some noises drawn from wrapped Gaussian distribution during the diffusion process. Then, we predict the added noises with the extracted atom representations corresponding to the torsional angle. In Table 9, we can see clear improvements on PIP and PSR tasks compared with GearNet-Edge. This suggests that it would be a promising direction to explore more different diffusion models for pre-training, *e.g.*, diffusion models on backbone dihedral angles.

# H    Results of Pre-Training GVP

Table 10: Atom-level results of GVP on Atom3D tasks. Accuracy is abbreviated as Acc.

| | Method | PSR | | MSP | PIP | RES | Mean Rank |
|---|---|---|---|---|---|---|---|
| | | Global $\rho$ | Mean $\rho$ | AUROC | AUROC | Acc. | |
| | GVP | 0.809 | 0.486 | 0.610 | 0.846 | 0.481 | 8.6 |
| w/ pre-training | Denoising Score Matching | 0.849 | 0.535 | 0.625 | 0.851 | 0.529 | 5.4 |
| | Residue Type Prediction | 0.845 | 0.527 | 0.652 | 0.847 | 0.518 | 5.8 |
| | Distance Prediction | 0.825 | 0.513 | 0.632 | 0.836 | 0.483 | 7.6 |
| | Angle Prediction | **0.872** | 0.545 | 0.637 | **0.881** | **0.557** | **2.0** |
| | Dihedral Prediction | 0.852 | 0.538 | **0.677** | **0.881** | 0.555 | 2.2 |
| | Multiview Contrast | 0.848 | 0.518 | 0.656 | 0.833 | 0.490 | 6.4 |
| | **DiffPreT** | 0.850 | 0.540 | 0.631 | 0.851 | 0.542 | 4.4 |
| | **SiamDiff** | 0.854 | **0.548** | 0.673 | 0.863 | 0.554 | 2.2 |

To study the effect of our proposed pre-training methods on different backbone models, we show the pre-training results on GVP [41, 42] in this section.

**Setup.** GVP constructs atom graphs and adds vector channels for modeling equivariant features. The original design only includes atom types as node features, which makes pre-training tasks with residue type prediction very difficult to learn. To address this issue, we slightly modify its architecture to add the embedding of atom and corresponding residue types as atom features. Then, the default configurations in Jing et al. [42] are adopted. We construct an atom graph for each protein by drawing edges between atoms closer than 4.5Å. Each edge is featured with a 16-dimensional Gaussian RBF encoding of its Euclidean distance. We use five GVP layers and hidden representations with 16 vector and 100 scalar channels and use ReLU and identity for scalar and vector activation functions, respectively. The dropout rate is set as 0.1. The final atom representations are followed by two mean pooling layers to obtain residue and protein representations, respectively. All other hyperparameters for pre-training and downstream tasks are the same as those in App. D.

Table 11: Spearman correlation on GB1 with the 2-vs-rest split.

| Method | Spearmanr $\rho$ |
|---|---|
| CNN | 0.320 |
| ESM-1b | 0.550 |
| GearNet-Edge | 0.651 |
| w/ Multiview Contrast | 0.714 |
| w/ Residue Type Prediction | 0.697 |
| w/ Distance Prediction | 0.685 |
| w/ Angle Prediction | 0.686 |
| w/ Dihedral Prediction | 0.661 |
| **w/ SiamDiff** | **0.747** |

Table 12: Comparison between SiamDiff with random torsional perturbation and sampling from a rotamer library.

| Method | PIP | MSP | RES | PSR | |
|---|---|---|---|---|---|
| | AUROC | AUROC | Accuracy | Global $\rho$ | Mean $\rho$ |
| **SiamDiff** | **0.884** | **0.698** | **0.460** | 0.829 | **0.546** |
| w/ rotamer library | 0.877 | 0.631 | 0.449 | **0.834** | 0.521 |

**Experimental results.** The results are shown in Table 10. Among all pre-training methods, SiamDiff, angle prediction and dihedral prediction are the top three. This is different from what we observe in Table 1, where residue type and distance prediction are more competitive baselines. We hypothesize that this is because GearNet-Edge includes angle information in the encoder while GVP does not. Therefore, GVP will benefit more from pre-training methods with supervision on angles. Nevertheless, we find that SiamDiff is the only method that performs well on different backbone models.

# I   Results on Protein Engineering Task

To further prove the effectiveness of our proposed pre-training methods, we add experiments on the GB1 dataset from FLIP [16]. As this is a protein engineering task with mutated sequences, we assume that the backbone structure remains unchanged after mutation, to save costs in generating mutant structures. We only keep CA atoms in the wild type protein structure as the input to the encoder. We benchmark residue-level methods in Table 11 in the attached file, alongside CNN and ESM-1b baselines from the FLIP paper.

According to Table 11, we observe that modeling structural information is beneficial compared with using only sequential information, even under the assumption that all mutants share the same backbone structure. Among all pre-training methods, SiamDiff demonstrates the most significant improvements over the baseline, once again validating the effectiveness of our method.

# J   Effects of Approximate Conformer Generation

In SiamDiff, we introduce approixmate conformer generation mechanism by randomly rotating torsional angles. We hypothesize that highly realistic conformers are not vital for better representations. To confirm, an extra rebuttal experiment is performed in Table 12. Instead of random perturbation, we sample from a rotamer library [62] based on residue types and backbone angles. Table 12 shows random torsional perturbation still outperforms sampling from a rotamer library in most tasks, confirming our hypothesis. This can be attributed to the fact that the objective of pre-training is to learn common information between diverse views through mutual prediction, as SimCLR and SimSiam. Considering this perspective, introducing random torsional noise allows us to generate more diverse conformers compared to solely relying on realistic conformer distributions.

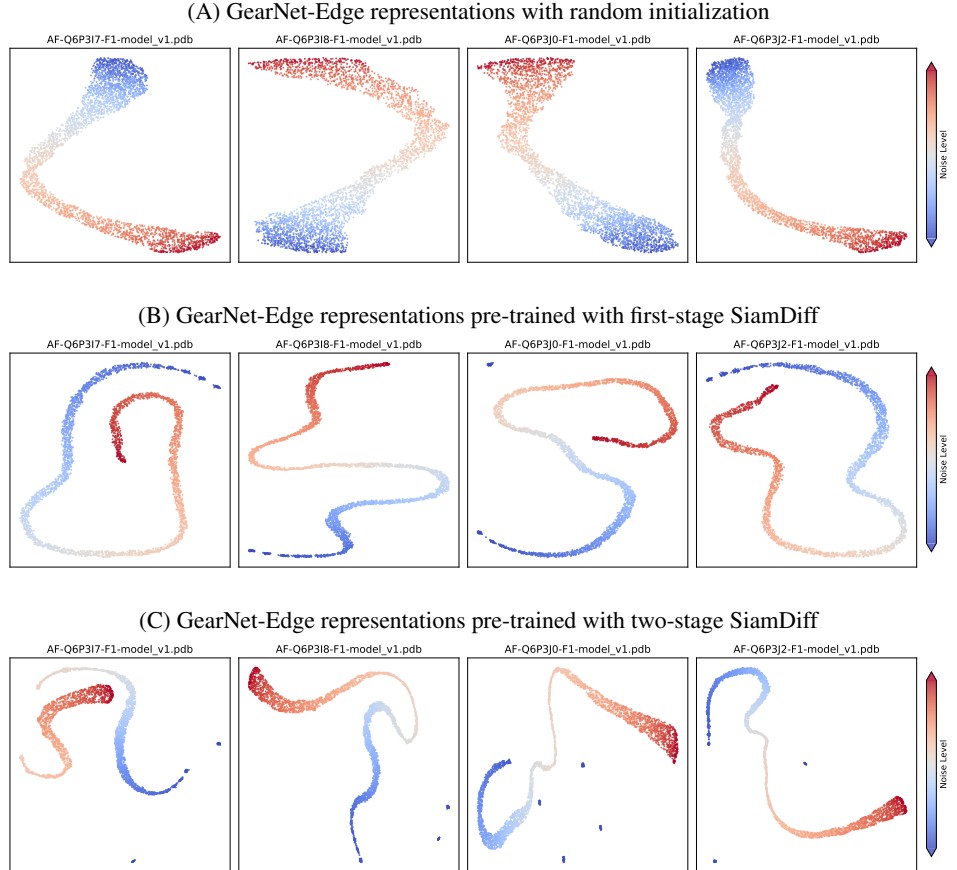

Figure 4: Visualization of GearNet-Edge representations on 4 randomly selected proteins from AlphaFold Database with sequence and structure noises at different levels. Three models are employed for extracting representations: (A) without pre-training, (B) with first-stage SiamDiff, and (C) with two-stage SiamDiff pre-training. For each protein, we consider $t = 1, ..., 100$ noise levels and randomly generate 32 noisy samples for each noise level using the scheme in SiamDiff. Samples with small noise levels are highlighted in blue, while those with large noise levels are marked in red.

## K    Visualization of SiamDiff Embeddings upon Noises

To explore pre-training insights, we visualize UMAP representations of 4 random AlphaFold DB proteins in Fig. 4. Several interesting phenomena can be observed:

- *Randomly initiated representations* in Fig. 4A form a clear, continuously color-changing trajectory (blue to red). This confirms that the forward diffusion process gradually adds noise to proteins, leading to smooth changes in their representations, as expected for diffusion models.

- *After pre-training with large noise scales*, the encoder maintains the color smoothness of the trajectory, which is desired for effective denoising during the backward diffusion process. Intriguingly, pre-training narrows the trajectory compared to the broader trajectory without pre-training, particularly at the two ends. This suggests that first-stage pre-training clusters proteins with similar levels of added noise, even for large and diverse noises. This clustering property proves useful for detecting large perturbations in downstream tasks, such as mutation stability prediction in MSP, as opposed to the diverse representation distributions in Fig. 4A.

- *Continuing with small noise scale pre-training*, the trajectory becomes much narrower in the middle and even breaks for some proteins. This indicates that by focusing on only slightly perturbed samples during pre-training, our model becomes capable of discerning proteins with small and large noises, making it more effective for fine-grained downstream tasks like PSR and

PIP. However, the red end of the trajectory is thicker than that in Fig. 4B, which may imply some forgetting behavior in the second-stage pre-training.

