# OpenReview forum: "Pre-Training Protein Encoder via Siamese Sequence-Structure Diffusion Trajectory Prediction"
_NeurIPS.cc/2023/Conference — NeurIPS 2023 spotlight_

### Official Review · Reviewer_E714 · 2023-06-27

**Soundness:** 3 good
**Presentation:** 2 fair
**Contribution:** 3 good
**Rating:** 6
**Confidence:** 3

**Summary:**

The authors propose to use the diffusion on both the protein sequence and the protein structure for pretraining. Additionally, they also take the correlation between different conformers of the same protein into consideration and maximize the mutual information between their trajectories via mutual denoising. Specifically, different conformers of the same protein are generated by perturbations on the side-chain torsional angles. Experiments on Enzyme Commission classification and four Atom3D tasks demonstrate the benefit of diffusion-based pretraining and mutual denoising.

**Strengths:**

1. The idea of maximizing the mutual information between diffusion trajectories of different conformers is interesting and inspiring. The authors also show the objective can be transformed into a loss function for mutual denoising between two relevant trajectories.
2. The evaluation is comprehensive. The authors compare with different pretraining strategy and conduct many ablation studies. They also explore the effect of the size of the pretraining data and different diffusion models.

**Weaknesses:**

1. Random perturbations on the side-chain torsional angles may produce conformations with high energy, which does not reflect the actual "physics underlying the conformational change" as claimed in line 51-52. This may also downgrade the effect of the mutual denosing scheme. As shown in the ablation studies, discarding the mutual information maximization scheme only has very minor effect on the performance.
2. The organization of the paper can be improved. The main insight lies in how to maximize the mutual information between representations of Siamese Trajectories. However, the major illustration of this contribution is put into the Appendix while the authors spend a great deal of time introducing the preliminaries on diffusion.

**Questions:**

1. Do you think the mutual denoising scheme can also work on homologs as you have mentioned the problem in line 50? There might not be a 1-to-1 mapping on the node representations when it comes to homologs since they may have different number of residues.
2. What do you think is the reason that diffusion-based pretraining produces better results than denoising score matching pretraining in your experiments? In Section 4.4, you have mentioned that perturbing distance matrices may produce negative values, however, it is also natural to do denoising score matching pretraining directly on the absolute coordinates [1][2]. Could the denosing score mathching baseline be underestimated because it is implemented on the distance matrix instead of the absolute coordinates?
3. Can you theoretically show how much deviation will be introduced by the approximation in line 743 (Appendix C.1)? For example, can you derive the bound of the gap between the lowerbound of mutual information and the loss you derived from the RHS of the approximation?
4. What is the value of $t$ in the diffusion model in the fine-tuning phase?

[1] Pre-training via Denoising for Molecular Property Prediction

[2] Energy-Motivated Equivariant Pretraining for 3D Molecular Graphs

**Limitations:**

Random perturbations on the side-chain torsional angles may not conform to the actual distribution of the conformational change.

---

> ### Author Rebuttal · Authors · 2023-08-07
>
> Thanks for your insightful comments and golden suggestions! We respond to your concerns as below:
>
> >**Q1: The random side-chain perturbation does not reflect the actual "physics underlying the conformational change**
>
> Please see the global response for details.
>
> >**Q2: Discarding the mutual information maximization scheme only has very minor effect on the performance**
>
> We respectfully disagree with the reviewer on this point. It should be noted that the SiamDiff method without mutual information maximization scheme is exactly the DiffPreT method. As shown in Tables 1 & 2, the improvement of SiamDiff over DiffPreT is consistent and significant, especially on residue-level tasks.
>
> >**Q3: The organization of the paper can be improved. The major illustration of this contribution is put into the Appendix while the authors spend a great deal of time introducing the preliminaries on diffusion.**
>
> Thanks for the suggestion! In the revised version, we will move a portion of preliminaries of diffusion models to the appendix, thus leaving more space to discuss and evaluate the effect of different diffusion design choices and conduct more downstream evaluation. We believe this can be done in the camera ready version with an additional content page limit.
>
> >**Q4: Do you think the mutual denoising scheme can also work on homologs as you have mentioned the problem in line 50?**
>
> This is an interesting question! It is indeed possible to use structural homologs instead of randomly generated conformers for mutual denoising. Though there might not be one-to-one residue mapping, we can still use the alignment tools between protein structures [a,b] to get residue/node correspondence for mutual denoising. Nevertheless, extra efforts are required to study whether these realistic conformers are better for pre-training, as discussed in our response to Q1. We leave this question to future work.
>
> [a] Zhang, Yang, and Jeffrey Skolnick. "TM-align: a protein structure alignment algorithm based on the TM-score." Nucleic acids research 33.7 (2005): 2302-2309.
>
> [b] Holm, Liisa. "Using Dali for protein structure comparison." Structural Bioinformatics: Methods and Protocols (2020): 29-42.
>
> >**Q5: What do you think is the reason that diffusion-based pretraining produces better results than denoising score matching pretraining in your experiments? Could the denosing score mathching baseline be underestimated because it is implemented on the distance matrix instead of the absolute coordinates?**
>
> Regarding denoising score matching, we agree that it is more appropriate and intuitive to add noises directly to the absolute coordinates instead of distance matrices. In our ablation study, the baseline (SiamDiff w/o sequence diffusion) can be considered as an augmented version of denoising score matching. The superiority of SiamDiff over denoising score matching can be attributed to two main factors: sequence-structure joint diffusion and multi-level noise scheduling, both of which have been extensively discussed in our ablation study.
>
> Furthermore, as mentioned in Sec. 3.4, another drawback of previous denoising-based molecular pre-training methods is that they typically treat the noise level as a hyperparameter to be tuned [a]. This presents a significant challenge in selecting the optimal hyperparameter for pre-training, as it becomes difficult to capture both coarse- and fine-grained features effectively.
>
> [a] Zaidi, Sheheryar, et al. "Pre-training via denoising for molecular property prediction." ICLR, 2023.
>
> >**Q6: Can you theoretically show how much deviation will be introduced by the approximation in line 743 (Appendix C.1)? For example, can you derive the bound of the gap between the lower bound of mutual information and the loss you derived from the RHS of the approximation?**
>
> Thanks for your careful reading and insightful question! The approximation made in line 743 is based on replacing representations with diffusion trajectories. The latter is easier to sample and provides more informative targets for recovery. Therefore, this decision is more of a practical choice. In theory, it would be possible to derive a lower bound for the error introduced in this approximation by introducing some assumptions about the information loss between representations and trajectories. However, since the primary focus of the paper lies in the application of the pre-training algorithm, the presented theorems serve as a justification for the reasonableness of our method and only consider rough approximations. Characterizing theoretically the deviation introduced by such approximations is a very interesting direction for future work.
>
> >**Q7: What is the value of $t$ in the diffusion model in the fine-tuning phase?**
>
> We want to clarify that we only introduce noises in the pre-training stage and will use the original protein in the fine-tuning stage, which is common in denoising pre-training methods [a,b]. The aim of pre-training is to make the learned representations capable of capturing structural and sequential details. We don’t need to introduce additional noise for downstream tasks. We will clarify this point in the revised version.
>
> [a] Zaidi, Sheheryar, et al. "Pre-training via denoising for molecular property prediction." ICLR, 2023.
>
> [b] Liu, Shengchao, Hongyu Guo, and Jian Tang. "Molecular geometry pretraining with se (3)-invariant denoising distance matching." ICLR, 2023.

---

> > ### Comment · Reviewer_E714 · 2023-08-13
> > **Thanks for your response**
> >
> > Thanks for your detailed response, which largely alleviates my concerns. Thus I raised my rating.

---

> > > ### Author Response · Authors · 2023-08-15
> > >
> > > Thank you for your suggestions and response! We'll follow your suggestions to continue to work on the revision of the paper.

---

### Official Review · Reviewer_Qtp8 · 2023-06-28

**Soundness:** 3 good
**Presentation:** 2 fair
**Contribution:** 3 good
**Rating:** 7
**Confidence:** 3

**Summary:**

The authors propose a novel pre-training method for proteins by jointly modeling sequences and structures with a diffusion model (DiffPreT). The encoder, which is the noise prediction network, learns representations for the sequence and the structure, respectively. This representation can then be used for downstream tasks. Additionally, the authors sugest an extension to their method to capture the correlation between conformer structures, based on the maximization of mutual information between diffusion trajectories for different simulated conformers, obtained by perturbing side chain rotations.

**Strengths:**

The authors bring forward some very interesting ideas and the results are promising, possibly signifying a step forwards within the field of protein representations. Moreover, it is clear that an extensive amount of work has gone into running experiments, including ablation studies and additional results in the appendix.

**Weaknesses:**

The paper is confusing and unclear in some places, both in terms of text and in terms of mathematical notation. There are some essential parts of the method that are not properly explained and/or motivated, and not all results seem to indicate a significant improvement. Finally, I think the results could be presented in a more diverse way, rather than only showing big tables.

For comments and suggestions on how to imrpove the manuscript, I refer to the "Questions" part.

**Questions:**

- Main questions / comments:
  1. In the DiffPreT setting (and to some extent also in the SiamDiff setting), it is unclear to me what noise level is used to get the representation for downstream tasks. It is mentioned that mutiple levels of noise are used, but it remains vague wether this means concatenating respresentations at multiple noise levels, all noise levels, or picking a noise level at random when getting the representation for the downstream task.
  2. Maybe even more generally, could you give some intuition on why representations on noisy structures could serve better for downstream tasks?
  3. Could you comment on the size of the latent representations that you get? I might have overlooked it, but I missed what the size of $d$ is. This seems like important information, especially when the SiamDiff setting is used where all these representations seem to be concatenated.
  4. For the siamDiff setting, it is not intuitive to me what kind of distribution is really being modelled. Randomly rotating the side chains without perturbing the backbone, and simply throwing away structures where there are clashes seems a bit messy. Is the resulting something "biologically relevant"?
  5. When considering the standard deviation of some of the results, the improvements are not always statistically significant. Perhaps it would be good to acknowledge this.
  6. The structuring of the paper is a bit off in some places, e.g. 3.4 is showing results even before all methods are explained, and the discussion in 4.4 is more a "related works" section, which has mostly been discussed in the introduction already, and which somewhat disrupts the flow of the paper.
  7. From the main text, it took me a long time to discover that the encoder $\phi$ is the noise predicting network, and even longer to see that it is GearNet-Edge (and GVP in the appendix). This could be more clear earlier on in the manuscript.
  8. In the appendix it is stated that GearnNet-Edge is a structural encoder. However, from the main paper it seems like you get a representation for both the sequence and the structure. How does this work? Are these separate networks?
  9. All results are presented as big tables, it would help to include some intuitive graphical results. Perhaps visualize the protein structure / sequence reconstructions of the diffusion model, or somehow visualize the representations (for example see if clusters appear when PCA is done on the representations or something similar).


- Other questions / comments:
  10. The term "encoder" can be confusing, as the forward process of a diffusion model can also be seen as a series of unparameterized encoders. It would help if this distinction is clear from the beginning.
  11. Perhaps a missing reference for protein structure-sequence co-design: Lisanza et al. [2023]
  12. The task explanation in the main paper is very minimal. I appreciate that there is a more extensive description in the appendix, but it could still be much more elaborate.
  13. Sometimes abbreviations are introduced before their meaning (e.g. EC, ESM).
  14. The "Mean Rank" seems like a strange metric to me that does not add much to all other results.
  15. The results using GVP are interesting and would maybe be worth including in the main paper, including a discussion on why some scores are much higher than for GearNet-Edge (except for PSR).
  16. Some discussion on computational cost is missing, either in the main paper or in the appendix.

**Limitations:**

The authors discuss limitations in the appendix. It would be nice if some of this discussion is transferred to the main paper.

---

> ### Author Rebuttal · Authors · 2023-08-07
>
> Thanks for your careful reading! While some questions may be due to misunderstanding, we find that your suggestions are very helpful for us to improve the quality and clarity of our paper! We respond to your concerns below:
>
> >**Q1: What noise level is used to get the representation for downstream tasks? Why noisy protein representations can help?**
>
> We clarify that we only introduce noises during pre-training and will use the original protein during fine-tuning, as in denoising pre-training [a,b]. The aim of pre-training is to make the learned representations capture structural and sequential details. We don’t need to introduce additional noise for downstream tasks. We will clarify this point in the revised version.
>
> [a] Zaidi et al. "Pre-training via denoising for molecular property prediction." ICLR, 2023.
>
> [b] Liu et al. "Molecular geometry pretraining with se (3)-invariant denoising distance matching." ICLR, 2023.
>
> >**Q2: Please comment on the size of the latent representation that you use.**
>
> As stated in the Appendix D, our approach employs 128 hidden dimensions for atom-level protein structure tasks and 512 for residue-level tasks. We use a smaller number of hidden dimensions in the atom-level cases so as to have moderate computational cost under the larger-scale atom-level graphs.
>
> >**Q3: Are the performance improvements on downstream tasks statistically significant?**
>
> Thanks for the question. To validate whether the improvements are statistically significant, we conduct a one-tailed t-test between our methods and the second best pre-training baselines on tasks that our method is the best at. We repeat the experiment with five different seeds and show the results in Table A.
>
> Table A: Statistical significance test results.
> |#Task|#Method|p-value|t-statistics|
> |:----:|:----:|:----:|:----:|
> |PIP (atom)|SiamDiff v.s. Residue Type Prediction|0.001|6.78|
> |RES (atom)|SiamDiff v.s. Residue Type Prediction|$5.0 \times 10^{-5}$|15.54|
> |PSR mean $\rho$ (atom)|SiamDiff v.s. Distance Prediction|0.003|5.11|
> |MSP (atom)|SiamDiff v.s. Distance Prediction|0.04|2.33|
> |EC auprc (residue)|SiamDiff v.s. Multiview Contrast|>0.1|-|
> |PSR global $\rho$ (residue)|SiamDiff v.s. Residue Type Prediction|0.01|3.68|
> |PSR mean $\rho$ (residue)|SiamDiff v.s. Residue Type Prediction|0.01|3.62|
>
> It can be observed that for all tasks except EC, the performance improvements are statistically significant under a p-value less than 0.1 (i.e., with t-statistics surpassing the critical value of the corresponding test). We will include the test results in the camera ready version and acknowledge that the improvements on EC are not statistically significant.
>
> >**Q4: Are the conformers derived by side chain perturbation “biologically relevant”?**
>
> Please see the global response for details.
>
> >**Q5: Sec. 3.4 is showing results even before all methods are explained.**
>
> For Sec. 3.4, we want to emphasize the contribution of the proposed two-stage noising scheme as a part of our method and **the results are necessary to support our claims in Sec. 3.4**. First, we provide intuitive insights into the challenges of denoising sequence and structure with varying noise levels, emphasizing the advantages of two-stage noise scheduling. To validate these points, we examine structure denoising loss and sequence denoising accuracy across diverse noise levels and schedules, effectively showcasing our two-stage noise scheduling approach.
>
> >**Q6: How can you get both sequence and structure representations using GearNet-Edge?**
>
> In the paper, we have mentioned in the first footnote that the structure encoder refers to those that take both sequences and structures as input. Notably, GearNet-Edge employs sequential edges, linking consecutive residues in a protein sequence. In this way, message passing is performed according to the protein sequence, and thus GearNet-Edge can extract sequence representations. In another way, spatial and KNN edges are constructed to represent the protein structure, which enables GearNet-Edge to extract structure representations. By combining these different types of edges, GearNet-Edge extracts both sequence and structure representations.
>
> >**Q7: It would help to include some intuitive graphical results.**
>
> Please see the global response for details.
>
> >**Q8: A related work of protein structure-sequence co-design [a] is not referred to.**
>
> This work is closely related to ours in terms of performing sequence-structure joint diffusion. However, our work focuses on learning informative protein representations, while this work focuses on protein generation. We will discuss these connections and differences in the revision.
>
> [a] Lisanza, et al. "Joint generation of protein sequence and structure with RoseTTAFold sequence space diffusion." bioRxiv, 2023.
>
> >**Q9: The metric “Mean Rank” does not add much to all other results.**
>
> We would like to argue that **“Mean Rank” is a commonly used metric for benchmarking methods on a diverse set of tasks**. As shown in Tables 1 and 2, some baseline methods perform well (even the best) on one or two tasks while cannot perform consistently well on all tasks. Since it is hard to measure such performance consistency using individual task-specific metrics, we introduce “Mean Rank'' as a metric for measuring performance consistency across tasks. Based on this metric, we can observe that DiffPreT and SiamDiff are the only two methods that perform consistently well on all tasks.
>
> >**Q10: The discussion on computational cost is missing.**
>
> Regarding computational cost, the protein encoder remains the main bottleneck, involving linear/quadratic cost due to message passing among atoms in the graph.  Perturbing/recovering protein sequences and structures carries linear complexity relative to atoms/residues, significantly cheaper than the protein encoder. We will add a paragraph in the method section to discuss the computational cost introduced in pre-training.

---

> > ### Comment · Reviewer_Qtp8 · 2023-08-15
> >
> > First of all, I would like to thank the authors for their detailed rebuttal and the work they put into generating new, interesting results. All my comments were addressed in a concise manner, and all newly reported results and visualizations add to the value of the paper. Given that the authors will improve the clarity of the paper to avoid misunderstandings for future readers, I will happily recommend this paper to get accepted and increase my score (5 $\rightarrow$ 7).
> >
> > One additional question out of curiosity: I really like the discussion you provided for Figure 4 of the new results, can you also discuss the low-noise outliers (i.e. the small dark blue islands)?

---

> > > ### Author Response · Authors · 2023-08-15
> > >
> > > Thanks for your resopnse! We'll follow your suggestions to revise our paper in the final version.
> > >
> > > The presence of low-noise outliers first becomes noticeable in Fig. 4(B) and is more evident in Fig. 4(C). These outliers represent proteins with minimal noise. Through pre-training on small noise, our model successfully differentiates between proteins without noise and those with minimal noise. This implies that our model can detect even subtle perturbations in the protein, an observation that might seem intuitive given the number of masks within the protein. Nonetheless, further investigation in higher dimension is needed to understand why several low-noise outliers appear, which we will continue to work on.

---

### Official Review · Reviewer_s72W · 2023-07-01

**Soundness:** 3 good
**Presentation:** 3 good
**Contribution:** 3 good
**Rating:** 7
**Confidence:** 3

**Summary:**

In this work, the authors perform a thorough investigation of different pre-training strategies on joint sequence-structure diffusion models for representation learning, rather than generative modeling. They evaluate on an EC prediction task and four tasks from the Atom3D benchmark. They find that joint sequence-structure pre-training with the SiamDiff method for maximizing mutual information between diffusion trajectories of protein conformers consistently outperforms other methods on downstream task evaluation.

**Strengths:**

Great observation that prior pre-training strategies excel for particular tasks and have shortcomings in others; and while SiamDiff does not yield huge improvements over other strategies, it does consistently outperform them across tasks.

This work takes a thorough and critical look at the benefits and tradeoffs of considering sequence- or structure-only methods, and systematically evaluates different pre-training approaches.


**Weaknesses:**

Lack of evaluation against other downstream predictors and backbone models for downstream prediction.

**Questions:**

The sizable gap in sequence denoising accuracy for sequence vs joint diffusion is very interesting, especially considering that in the ablation, the model without structure diffusion is competitive with the others. Can the authors comment on the additional complexity of joint diffusion compared to sequence-only diffusion, and why this accuracy gap is so pronounced? The authors mention model capacity, but clearly capacity is sufficient for downstream tasks.

**Limitations:**

Yes

---

> ### Author Rebuttal · Authors · 2023-08-07
>
> Thanks for your insightful comments and great suggestions! We respond to your concerns  below:
>
> >**Q1: Lack of evaluation against other downstream predictors and backbone models for downstream prediction.**
>
> **We would like to argue that the focus of the paper is the development and evaluation of new pre-training algorithms, not achieving state-of-the-art performance on a specific task.** Therefore, we constantly use GearNet-Edge as the backbone model and 2-layer MLP as the prediction head so as to **have a fair comparison among different pre-training methods**. We also try GVP as the backbone model in Appendix H, to show that our methods can be applied on different encoders. Indeed, it is interesting to study the performance of different pre-training methods under other backbone models like ProNet [a] and CDConv [b]. Since the response period is short, we will try our best to finish these studies and add them to the future paper version.
>
> [a] Wang, Limei, et al. "Learning hierarchical protein representations via complete 3d graph networks." ICLR, 2022.
>
> [b] Fan, Hehe, et al. "Continuous-Discrete Convolution for Geometry-Sequence Modeling in Proteins." ICLR, 2022.
>
> >**Q2: In the ablation, the model without structure diffusion is competitive with the others. What’s the additional complexity of joint diffusion against sequence-only diffusion?**
>
> First, we want to argue that structure diffusion is an important component of SiamDiff. According to Table 3, the baseline without structure diffusion does not perform well on structure-informed tasks like PIP, MSP and PSR. We also perform ablation studies on residue-level tasks during rebuttal, the results of which are shown in Table A. Based on these results, we can conclude that structure diffusion brings consistent improvements across all considered tasks, which proves its necessity.
>
> Table A: Ablation study on residue-level tasks.
> |#Method|EC||MSP|PSR||
> |:----:|:----:|:----:|:----:|:----:|:----:|
> ||AUPR|$F_{\max}$|AUROC|Global $\rho$|Mean $\rho$|
> |**SiamDiff**|**0.878**|**0.857**|**0.700**|**0.856**|**0.521**|
> |w/o structure diffusion|0.868|0.850|0.671|0.826|0.502|
>
> In terms of complexity, it should be noted that the bottleneck of computation still lies in the protein encoder. The encoder requires message passing between protein atoms and introduces linear (or quadratic) cost with respect to the number of edges in the protein graph. The process of perturbing and recovering protein structures only requires linear complexity with respect to the number of atoms/residues in the protein, which is similar to sequence diffusion and is cheap compared with the protein encoder. Therefore, it is good to include structure diffusion considering its benefits and little cost.
>
>
> >**Q3: Why is the gap of sequence denoising accuracy between joint diffusion and sequence-only diffusion so pronounced?**
>
> Thanks for the question! It is an interesting observation that sequence diffusion only can achieve around 0.8 recovery accuracy, while using joint diffusion will decrease the accuracy to 0.4. As explained in the paper, the phenomenon can be attributed to the introduction of large structural noise. The large effect of structural noise implies that the sequence recovery may rely on the backbone conformation of predicted residues (note that we remove the corresponding side chain atoms to avoid information leakage). After perturbing the backbone structures, it will be much more difficult to infer the residue types. This hypothesis can be validated by the phenomenon observed on residue-level pre-training: when using residue-level SiamDiff, we only keep the alpha Carbon atoms instead of all three backbone atoms. The sequence diffusion can only achieve around 0.4 recovery, while the joint diffusion will decrease the accuracy to 0.3, which has a much smaller gap. Therefore, we conclude that the large gap is because atom-level sequence denoising is *too easy* with the correct backbone information, while introducing structural noise brings difficulty to the pre-training task.

---

> > ### Comment · Reviewer_s72W · 2023-08-11
> >
> > I have read the rebuttal and thoroughly enjoyed the response! I particularly appreciate the authors' point that "the focus of the paper is the development and evaluation of new pre-training algorithms, not achieving state-of-the-art performance on a specific task". This is a refreshing focus and the paper should not be penalized for this. Additionally, I found the discussion on the impacts of joint diffusion on sequence recovery to be clarifying. I will happily increase my score and recommend acceptance of the paper.

---

> > > ### Author Response · Authors · 2023-08-15
> > >
> > > Thank you for your positive feedback and support! We will incorporate a discussion regarding the effects of joint diffusion on sequence recovery in the final version. We would be grateful if you could consider increasing your score.

---

> > > > ### Comment · Reviewer_s72W · 2023-08-18
> > > >
> > > > I have increased my score to 7. Thank you again for a productive and enjoyable discussion period.

---

### Official Review · Reviewer_MBi3 · 2023-07-06

**Soundness:** 3 good
**Presentation:** 4 excellent
**Contribution:** 3 good
**Rating:** 7
**Confidence:** 5

**Summary:**

The paper proposes to use joint protein sequence and structure diffusion as a pretraining task, which they call DiffPreT. In order to account for the fact that proteins can exist as ensembles of conformers, the paper further proposes to generate pairs of conformers, use the diffusion forward process to corrupt them, and then train the reverse process across diffusion trajectories. They call this pretrained model SiamDiff. The paper then shows that DiffPreT and SiamDiff are effective on a panel of protein function prediction tasks.

**Strengths:**

The paper addresses the important problem of protein function prediction using novel and effective methods. Using diffusion as a pretraining task is novel, interesting, and apparently quite effective. The SiamDiff objective is creative and also empirically effective. The experiments are well-done and convincing. The writing is clear, and the paper is easy to follow. I appreciate the uncertainties on metrics and the attempt to benchmark across a wide variety of tasks. Using two-stage noise-scheduling is also an intuitive way to adapt a model framework originally meant for generation to pretraining.

**Weaknesses:**

### Major

In general, this is a very good paper that clearly describes a promising method for protein pretraining. However, it only evaluates on Atom3D and EC number. The paper would be much more significant if they evaluated GearNet on zero-shot fitness tasks such as those in ProteinGym, protein engineering tasks such as those in FLIP, and more general structure prediction tasks such as those used in CAFA.

In addition, the results would be more general if the paper considered different diffusion schemes. For structure generation, diffusion on [frames](https://arxiv.org/abs/2301.12485) or [angles](https://arxiv.org/abs/2209.15611) seems to outperform diffusion directly on atoms. In sequence diffusion, D3PM with an absorbing state isn't as computationally efficient as [autoregressive diffusion](https://arxiv.org/abs/2110.02037) while not allowing iterative refinement like D3PM with a uniform prior over amino acids. The paper would be slightly stronger if it discussed these choices, and much stronger if it evaluated the effect of these choices. However, I realize that this is probably not feasible during the discussion period!

In general, I think the exposition of the diffusion model could be shortened, as that is not the main contribution of the paper, in order to make more space for contributions that would increase the paper's significance.

### Minor

- The paper should include ablations on EC
- It's not clear to me why ESM-2-650M-GearNet doesn't require structure as input.

**Questions:**

- How is EC a residue-level task? Don't the labels apply to the entire protein?
- Is there a way to use this model on proteins without a structure?
- How does the choice of diffusion process effect performance?
- How well does this work on function prediction, zero-shot fitness prediction, or protein engineering tasks?

**Limitations:**

- SiamDiff requires structures at inference time.
- The paper only considers one diffusion process out of many possible choices.

---

> ### Author Rebuttal · Authors · 2023-08-07
>
> Thanks for your appreciation of our work! We respond to your questions and concerns below:
>
> >**Q1: The evaluation on more types of downstream tasks would make this paper more significant.**
>
> Thanks for the suggestion. We believe this additional experiment in the global response showcases the potential of SiamDiff on protein engineering tasks. However, we acknowledge that more effort is required when considering more complicated settings, e.g., insertion and deletion. Also, task-specific designs need to be explored to apply structure-based models for zero-shot fitness prediction. Due to the limited time in the rebuttal period, we leave these explorations as future work.
>
> >**Q2: Different structure and sequence diffusion schemes are not thoroughly explored in the current draft.**
>
> Thanks for pointing out this important aspect!
>
> For structure diffusion, we resort to **diffusion on coordinates** for its **effectiveness on learning molecular representations that has been proven in recent small molecule pre-training works [a,b]**. We agree that **diffusion on amino acid frames and bond/torsion angles better fit the inductive biases of protein structure generation** as shown in FoldingDiff [c] and RFdiffusion [d]. It is intriguing to study the effectiveness of these structure diffusion schemes on protein representation learning. We will definitely investigate them in our future work.
>
> For sequence diffusion, Autoregressive Diffusion Models (ARDMs) [e] are equivalent to the infinite time limit of Discrete Denoising Diffusion Probabilistic Models (D3PMs) [f] that are used in our current work. Therefore, **ARDMs are maximally expressive and can potentially enhance the effectiveness of our proposed DiffPreT and SiamDiff methods.** To explore this possibility, we conducted an initial experiment during the rebuttal by replacing D3PMs in SiamDiff with ARDMs. The results are presented in Table A.
>
> Table A: Comparison between SiamDiff with D3PMs and ARDMs.
> |#Method|PIP|MSP|RES|PSR||
> |:----:|:----:|:----:|:----:|:----:|:----:|
> ||AUROC|AUROC|Acc.|Global $\rho$|Mean $\rho$|
> |**SiamDiff**|**0.884**|**0.698**|**0.460**|**0.829**|**0.546**|
> |w/ ARDM|0.883|0.640|0.450|0.828|0.533|
>
> As shown in Table A, the results of ARDMs are quite close to those of SiamDiff on some tasks, but generally fall short in performance. Our hypothesis is that the advantages of ARDMs in the original paper stem from their random assigned order and causal masking in Transformers. The causal masking scheme poses challenges for GNN-based encoders. In our implementation, we simply remove edges that do not adhere to the assigned order. Further investigation is needed to explore how to adapt ARDMs to proteins, and this aspect is left for future work.
>
> [a] Zaidi, Sheheryar, et al. "Pre-training via denoising for molecular property prediction." ICLR, 2023.
>
> [b] Liu, Shengchao, Hongyu Guo, and Jian Tang. "Molecular geometry pretraining with se (3)-invariant denoising distance matching." ICLR, 2023.
>
> [c] Wu, Kevin Eric, et al. "Protein structure generation via folding diffusion." arXiv, 2022.
>
> [d] Watson, Joseph L., et al. "De novo design of protein structure and function with RFdiffusion." Nature, 2023.
>
> [e] Hoogeboom, Emiel, et al. "Autoregressive diffusion models." ICLR, 2022.
>
> [f] Austin, Jacob, et al. "Structured denoising diffusion models in discrete state-spaces." NeurIPS, 2021.
>
> >**Q3: The paper should include ablations on EC.**
>
> Thanks for the suggestion. We agree that including the ablation on EC will provide a better understanding of components of SiamDiff on residue-level tasks. However, this is infeasible during the rebuttal period, due to the large amount of computational resources needed (5 pre-training baselines * 3 repeated experiments, each with 4 GPUs and 24 hours fine-tuning). We will run the experiments and add the ablation study in the final version.
>
> >**Q4: It’s not clear why ESM-2-650M-GearNet doesn’t require structure as input.**
>
> We would like to clarify that what we state in the paper is ESM-2-650M (the sequence encoder part of ESM-2-650M-GearNet) can only extract representations from protein sequences and cannot take protein structures as input, making it unsuitable for structure-related tasks in Atom3D. **ESM-2-650M-GearNet can definitely take protein structures as input and extract structural representations by its GearNet component.** We will make this point clearer in the revision.
>
> >**Q5: How is EC a residue-level task? Don’t the labels apply to the entire protein?**
>
> We clarify that the “residue-level” here means that **residue-level structures are used for task prediction, instead of predicting per-residue labels**. Therefore, the labels of the EC task are applied to entire proteins, and we use residue-level protein structures to predict such per-protein labels. We will state this point more clearly in the revised version.
>
> >**Q6: Is there a way to use this model on proteins without a structure? SiamDiff requires structures at inference time.**
>
> As joint pre-training methods upon protein sequences and structures, DiffPreT and SiamDiff require protein structures as input for model prediction. Given the recent advancement in protein structure prediction, it can be easy to obtain accurate structures for most proteins, though with additional inference time. However, we argue that in most protein-related tasks, the importance of accuracy outweighs that of inference speed, including protein function and protein-protein interaction prediction. Besides, even for high-throughput tasks dominated by sequence-based methods like protein engineering, structure-based methods can be easily applied with little extra effort and large improvements, as shown in the GB1 experiment in the global response. Hence, we argue that the requirement of protein structures should not be a downside of our method.

---

> > ### Comment · Reviewer_MBi3 · 2023-08-10
> > **Thank you for the rebuttal**
> >
> > I am impressed with the additional experiments the authors managed to do during the rebuttal period and think that they increase the contribution of the paper. However, on GB1 2-vs-rest, there are some stronger baseline results in Table 3 here: https://www.biorxiv.org/content/10.1101/2022.05.19.492714v4.full.pdf
> >
> > In general, I think this is a strong and interesting paper that deserves acceptance.

---

> > > ### Author Response · Authors · 2023-08-11
> > > **Thank you for your response**
> > >
> > > Thanks for bringing our attention to that work! We'll include that baseline in our final version. We acknowledge that these protein engineering tasks are very interesting and promising to study. We'll work on that with structure-based methods in the future.

---

### Author Rebuttal · Authors · 2023-08-07

We extend our gratitude to all reviewers for valuable feedback. We’ve made significant improvements based on your suggestions. Here is a brief summary of important points:

>**New benchmark results on protein engineering task (Reviewer MBi3)**

During the rebuttal period, we followed your suggestion to include the GB1 dataset from FLIP in experiments. As this is a protein engineering task with mutated sequences, we assume that *the backbone structure remains unchanged after mutation*, to save costs in generating mutant structures. We only keep CA atoms in the wild type protein structure as the input to the encoder. We benchmark residue-level methods in Table 11 in the attached file, alongside CNN and ESM-1b baselines from the FLIP paper.

According to Table 11, we observe that modeling structural information is beneficial compared with using only sequential information, even under the assumption that all mutants share the same backbone structure. **Among all pre-training methods, SiamDiff demonstrates the most significant improvements over the baseline, once again validating the effectiveness of our method.**

>**Biological relevance of random torsional perturbation scheme (Reviewer Qtp8, E714)**

First, we reiterate that random side-chain perturbation is commonly used for simulating conformers [a,b]. Moreover, removing structures with clashes ensures that generated conformers are physically plausible. So the generated conformer distribution is biologically relevant.

Methodologically, while adding torsional noises without changing the backbone when generating conformers, Gaussian noises are introduced to the backbone during forward diffusion. This makes our encoder capture both backbones and side-chain noises effectively.

Besides, in pre-training, highly realistic conformers aren’t vital for better representations. To confirm, an extra rebuttal experiment (Table 12 in attached file) is performed. Instead of random perturbation, we sample from a rotamer library [c] based on residue types and backbone angles. Table 12 shows random torsional perturbation still outperforms sampling from a rotamer library in most tasks, confirming our hypothesis. This can be attributed to the fact that the objective of pre-training is to learn common information between diverse views through mutual prediction, as SimCLR and SimSiam. Considering this perspective, introducing random torsional noise allows us to generate more diverse conformers compared to solely relying on realistic conformer distributions.

In summary, while random torsional perturbation may not be as realistic as rotamer library-based or force field-based methods, it holds **biological relevance, is easy to implement, and proves to be a practical pre-training choice due to performance advantages**.

[a] Ho et al. "Probing the flexibility of large conformational changes in protein structures through local perturbations." PLoS computational biology, 2009.

[b] Ho et al. "Conserved tertiary couplings stabilize elements in the PDZ fold, leading to characteristic patterns of domain conformational flexibility." Protein Science, 2010.

[c] Shapovalov et al. "A smoothed backbone-dependent rotamer library for proteins derived from adaptive kernel density estimates and regressions." Structure, 2011.

>**New visualization results (Reviewer Qtp8)**

We have added visualization results in the attached pdf file. To explore pre-training insights, we visualize UMAP representations of 4 random AlphaFold DB proteins in Fig. 4.

Several interesting phenomena can be observed:
1. *Randomly initiated representations* in Fig. 4(A) form a clear, continuously color-changing trajectory (blue to red). **This confirms that the forward diffusion process gradually adds noise to proteins, leading to smooth changes in their representations, as expected for diffusion models.**
2. After *pre-training with large noise scales*, the encoder maintains the color smoothness of the trajectory, which is desired for effective denoising during the backward diffusion process. Intriguingly, pre-training narrows the trajectory compared to the broader trajectory without pre-training, particularly at the two ends. **This suggests that first-stage pre-training clusters proteins with similar levels of added noise, even for large and diverse noises.** This clustering property proves useful for detecting large perturbations in downstream tasks, such as mutation stability prediction in MSP, as opposed to the diverse representation distributions in Fig. 4(A).
3. Continuing with *small noise scale pre-training*, the trajectory becomes much narrower in the middle and even breaks for some proteins. **This indicates that by focusing on only slightly perturbed samples during pre-training, our model becomes capable of discerning proteins with small and large noises, making it more effective for fine-grained downstream tasks like PSR and PIP.** However,  the red end of the trajectory is thicker than that in Fig. 4(B), which may imply some forgetting behavior in the second-stage pre-training.

>**Paper presentation (Reviewer MBi3, Qtp8, E714)**

We acknowledge the concerns raised by the reviewers regarding the paper's presentation. Due to the extensive methodology and experimental contribution, it was challenging to fit everything within the 9-page limit. In the submitted version, we dedicated a lengthy section to explaining the fundamental concepts, making it more accessible to readers unfamiliar with diffusion models on proteins. However, we recognize that many intriguing observations and experiments were relegated to the appendix. We believe this problem can be addressed in the camera ready version with an additional page limit. We plan to reorganize our paper based on your suggestions, including the shortening of Sec. 3 and the addition of more details in Sec. 4, merging Sec. 4.4 into the Related Work section, and providing necessary descriptions about the tasks and experimental settings.

---

### Author Response · Authors · 2023-08-15
**Thanks for all the reviewers' suggestions and feedback!**

We'd like to extend our gratitude to all the reviewers for their insightful questions, prompt responses, and constructive feedback. We truly value our discussions with you. Your suggestions are helpful for us to add more interesting experiments and discussions to the paper. We will continue to follow your suggestions on revising the manuscript.

---

### Decision · Program_Chairs · 2023-09-21

**Decision:**

Accept (spotlight)

**Comment:**

The paper propose to pretrain neural models for predicting properties of proteins from sequences by maximizing mutual information between denoising trajectories of pairs of related conformers. The authors lay out the diffusion model first, which constituted DiffPreT, and then propose the trajectory based maximum mutual information objective, called SimDiff. They show this objective can be optimized by optimizing a lower bound on it.

The reviewers found the paper quite compelling. Some reviewers raise the point that maybe in some case the results were not SOTA, but they concurred with the authors' contention that goal of the paper was to show that the approach results in a useful pre-training strategy, that produces strong gains in several cases.

I felt that the authors' did a really good job of answering the reviewer's criticisms. As a result several of the reviewers raised their original assessment and were quite positive on the paper.